# Rethinking Climate, Climate Change, and Their Relationship with Water

**Demetris Koutsoyiannis** 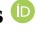

Department of Water Resources and Environmental Engineering, School of Civil Engineering, National Technical University of Athens, 15780 Athens, Greece; dk@itia.ntua.gr

**Abstract:** We revisit the notion of climate, along with its historical evolution, tracing the origin of the modern concerns about climate. The notion (and the scientific term) of climate was established during the Greek antiquity in a geographical context and it acquired its statistical content (average weather) in modern times after meteorological measurements had become common. Yet the modern definitions of climate are seriously affected by the wrong perception of the previous two centuries that climate should regularly be constant, unless an external agent acts upon it. Therefore, we attempt to give a more rigorous definition of climate, consistent with the modern body of stochastics. We illustrate the definition by real-world data, which also exemplify the large climatic variability. Given this variability, the term "climate change" turns out to be scientifically unjustified. Specifically, it is a pleonasm as climate, like weather, has been ever-changing. Indeed, a historical investigation reveals that the aim in using that term is not scientific but political. Within the political aims, water issues have been greatly promoted by projecting future catastrophes while reversing true roles and causality directions. For this reason, we provide arguments that water is the main element that drives climate, and not the opposite.

**Keywords:** climate; climate change; water; hydrology; climatology

*There's no such thing as bad climate, only bad clothes*

Paraphrasis of a Scandinavian proverb

*Each definition is a piece of secret ripped from Nature by the human spirit.*

Nikolai Luzin (from [1])

## 1. Introduction

As concerns about climate become all the more widespread in society, it is useful to revisit the notion of climate in scientific terms, along with its historical evolution and the very origin of the concerns per se. Water issues have been central among climate concerns, and thus it is useful to clarify the relationship between climate and water. As will be detailed in Section 2, the notion (and the scientific term) of climate was established in ancient Greece in a geographical context, while it acquired a statistical content (average weather) in modern times after meteorological measurements had become common. Yet the modern definitions of climate, which are discussed in Section 3, are deficient as they are affected by a wrong perception of the 19th and 20th centuries that the climate at a certain place should regularly be constant, unless an external agent acts upon it. For this reason, in Section 4 we attempt to propose a more rigorous and consistent definition of climate, based on stochastics. We illustrate the definition by real-world data, which also exemplify the large variability of climate. Given this variability, the term "climate change" turns out to be scientifically unjustified. Specifically, it is a pleonasm, as the climate, like the weather, has been ever-changing. Indeed, the analysis in Section 6 reveals that the objective of using the term is not scientific but political.

Concerning, in particular, the relationship of climate and water, the analysis of Section 5 shows that water is the main element that drives climate, rather than just being

affected by climate as commonly thought. This demands a more active role of hydrologists in climate research, replacing their current passive role in studying climate impacts.

To faithfully follow the development of ideas about climate from antiquity to modern science, we examine and quote several historical and modern texts. While the subject of this paper looks general, and its content perhaps trivial, the investigations performed, the information given and the synthesis thereof, are mostly new.

## 2. History of the Notion of Climate

Although the historian Herodotus (Ἡρόδοτος; c. 484–c. 425 BC) is perhaps the first who described different climates of some areas on Earth in a geographical context (see Appendix A), it was Aristotle (Figure 1) who, a century later, put the notion of climate in a scientific context. In his famous book *Meteorologica* he describes the climates on Earth in connection with latitude, but he uses a different term, *crasis* (κρᾶσις), literally meaning mixing, blending of things which form a compound, or temperament.

The term *climate* (κλίμα, plural κλίματα) was coined as a geographical term by the astronomer Hipparchus (Figure 1) in his Commentary on Aratus (Ἱππάρχου τῶν Ἀράτου καὶ Εὐδόξου φαινομένων ἐξηγήσεως [2]). Hipparchus is also known in climatology for his discovery and calculation of *precession of the equinoxes* (μετάπτωσις ἰσημεριῶν) by studying measurements on several stars. In the 20th century, this precession would be found to be related to the climate of the Earth and constitutes one of the so-called *Milankovitch cycles* (see Section 5). The term *climate* originates from the verb κλίνειν, meaning 'to incline' and originally denoted the angle of inclination of the celestial sphere and the terrestrial latitude characterized by this angle [2].

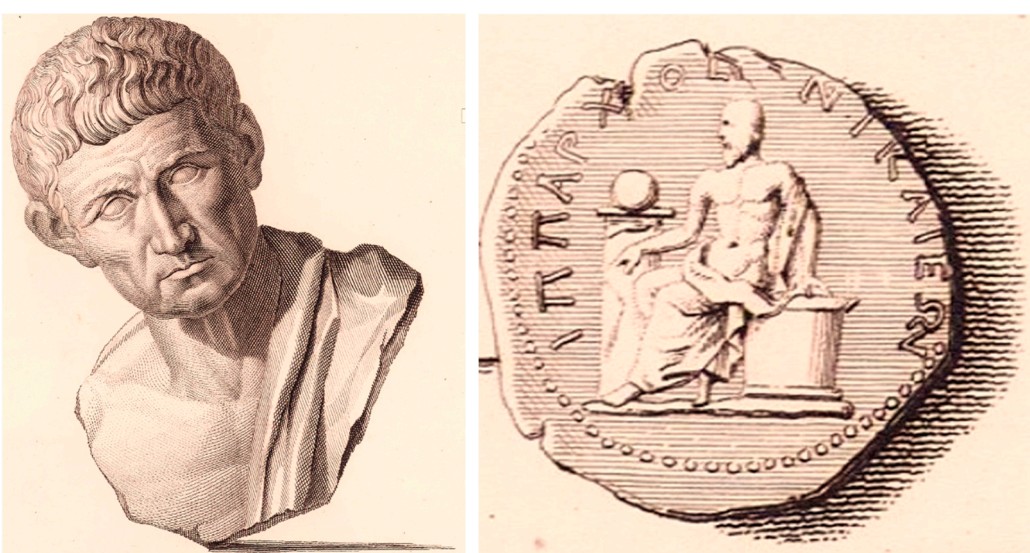

**Figure 1.** (**left**) Aristotle (Ἀριστοτέλης; 384–322 BC), Greek philosopher of the Classical period, founder of the Lyceum and the Peripatetic school of philosophy. (**right**) Hipparchus of Nicaea (Ἵππαρχος ὁ Νικαεύς; c. 190–c. 120 BC), Greek astronomer, geographer and mathematician founder of trigonometry and discoverer of the precession of the equinoxes, depicted in the back facet of a coin of the Roman period. (Image sources: [3]).

Hipparchus's *Table of Climates* was described by Strabo the Geographer (Στράβων; 64 or 63 BC–c. 24 AD), from whom it became clear that the *Climata* of that Table are just latitudes of several cities, from 16° to 58° N (for a reconstruction of the Table see Shcheglov [2]). However, Strabo himself used the term climate with a meaning close to the modern one. Furthermore Strabo, defined the five climatic zones, one *torrid*, two *temperate* and two *frigid*, as we use them to date (see also Appendix A).

The term climate was used with the ancient Greek geographical meaning until at least 1700, as imprinted in a dictionary of that era (see Appendix A). In contemporary times,

a search on old books [4] reveals that the term *climatology* appears after 1800. With the increased collection of meteorological measurements, the term *climate* acquired a statistical character as the average weather. Indeed, the geographer A.J. Herbertson (1907; [5]) in his book entitled "*Outlines of Physiography, an Introduction to the Study of the Earth*", gave the following definition of climate, based on, but also distinguishing it from, weather:

> By climate we mean the average weather as ascertained by many years' observations. Climate also takes into account the extreme weather experienced during that period. Climate is what on an average we may expect, weather is what we actually get.

Thus, Herbertson appears to be the father of the famous quotation "*climate is what we expect, weather is what we get*", often attributed to Mark Twain. (What Twain actually wrote, attributing it to an anonymous student, was "*Climate lasts all the time and weather only a few days*" [6]).

Herbertson also defined climatic regions of the world based on statistics of temperature and rainfall distribution, a work that was influential for the famous and most widely used climate classification by Köppen (1918; [7–11]). This includes six main zones and eleven climates which are on the same general scale as Herbertson's [12]. Herbertson's definition has been kept virtually without essential changes until now; for example, Lamb (1972; [13]) states:

> Climate is the sum total of the weather experienced at a place in the course of the year and over the years. It comprises not only those conditions that can obviously be described as 'near average' or 'normal' but also the extremes and all the variations.

A recent update of Earth's climate types is shown in Figure 2, while the distribution of land area, population and gross domestic product by climate zone is given in Table 1. It is evident from this table that humans have prosperously inhabited areas of virtually all diverse climates, thus vindicating the motto at the beginning of the paper that there is no such thing as bad climate, only bad clothes, where clothes here purports to be a metaphor for energy use by humans.

## 3. Modern Definitions of Climate

Since the early 20th century, Herbertson's definition of climate [5] was followed with slight amendments, which did not change the meaning. Here we quote a few of them referring to climate per se as well as with the tightly connected concept of the *climate system*:

(1)    By the USA National Weather Service [14]:

> Climate–The composite or generally prevailing weather conditions of a region, throughout the year, averaged over a series of years.

(2)    By the Climate Prediction Center of the National Weather Service [15]; notice that the Center refers to Herbertson's quotation as if it were an *old saying*:

> Climate–The average of weather over at least a 30-year period. Note that the climate taken over different periods of time (30 years, 1000 years) may be different. The old saying is climate is what we expect and weather is what we get.

(3)    By the American Meteorological Society [16]:

> Climate–The slowly varying aspects of the atmosphere–hydrosphere–land surface system. It is typically characterized in terms of suitable averages of the climate system over periods of a month or more, taking into consideration the variability in time of these averaged quantities. Climatic classifications include the spatial variation of these time-averaged variables. Beginning with the view of local climate as little more than the annual course of long-term averages of surface temperature and precipitation, the concept of climate has broadened and evolved in recent decades in response to the increased understanding of the underlying processes that determine climate and its variability.

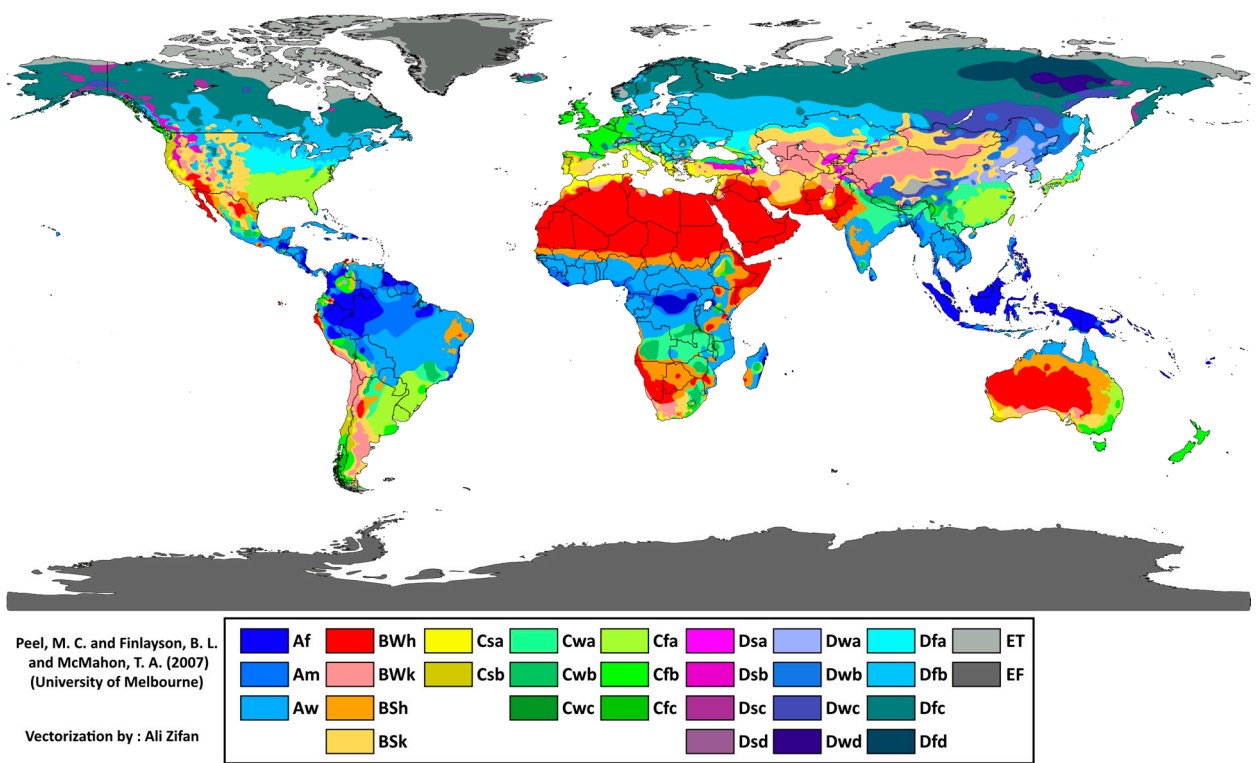

**Figure 2.** Updated world map of the Köppen climate classification, with the climate type groups appearing in the legend defined as: A-Tropical, B-Arid, C-Temperate, D-Cold, E-Polar (Source: [17,18]).

**Table 1.** Distribution of land area, population and gross domestic product (GDP) by climate zone (Source: [19]).

| Symbol | Description of Climate Type | % Area | % Population | % GDP |
|---|---|---|---|---|
| Af | Tropical rainforest | 4.0 | 4.4 | 2.8 |
| Am | Tropical monsoon | 0.8 | 2.4 | 1.0 |
| Aw | Tropical savanna | 10.8 | 17.5 | 6.6 |
| BW | Arid, desert | 17.3 | 6.2 | 3.6 |
| BS | Semi-arid (steppe) | 12.3 | 11.8 | 6.5 |
| Cs | Temperate with a dry summer | 2.2 | 4.3 | 9.1 |
| Cf | Temperate without a dry season | 7.7 | 19.5 | 43.7 |
| Cw | Temperate with a dry winter | 4.3 | 16.0 | 7.0 |
| Df | Continental without a dry season | 22.9 | 5.8 | 11.0 |
| Dw | Continental with a dry winter | 6.4 | 5.3 | 3.4 |
| E | Polar | 4.0 | 0.0 | - |
| H | Highland * | 7.3 | 6.8 | 5.3 |
| | Total | 100.0 | 100.0 | 100.0 |

* The Highland climate type, not being part of Köppen's original or revised classification depicted in Figure 2 but added by Finch and Trewartha [20], refers to highland areas (with altitude usually of more than 1500 m), namely the Cascades, Sierra Nevadas and Rockies of North America, the Andes of South America, the Himalayas and adjacent ranges and the Tibet Plateau of Asia, the eastern highlands of Africa, and the central portions of Borneo and New Guinea.

In turn, the American Meteorological Society defines the concept of the climate system as:

*The system, consisting of the atmosphere, hydrosphere, lithosphere, and biosphere, determining the earth's climate as the result of mutual interactions and responses to external influences (forcing). Physical, chemical, and biological processes are involved in the interactions among the components of the climate system.*

(4)  By the World Meteorological Organization (WMO) [21]:

*C0850 climate–Synthesis of weather conditions in a given area, characterized by long-term statistics (mean values, variances, probabilities of extreme values, etc.) of the meteorological elements in that area.*

*C0900 climate system–System consisting of the atmosphere, the hydrosphere (comprising the liquid water distributed on and beneath the Earth's surface, as well as the cryosphere, i.e., the snow and ice on and beneath the surface), the surface lithosphere (comprising the rock, soil and sediment of the Earth's surface), and the biosphere (comprising Earth's plant and animal life and man), which, under the effects of the solar radiation received by the Earth, determines the climate of the Earth. Although climate essentially relates to the varying states of the atmosphere only, the other parts of the climate system also have a significant role in forming climate, through their interactions with the atmosphere.*

(5)  By the Intergovernmental Panel on Climate Change (IPCC) [22]:

*Climate–Climate in a narrow sense is usually defined as the average weather, or more rigorously, as the statistical description in terms of the mean and variability of relevant quantities over a period of time ranging from months to thousands or millions of years. The classical period for averaging these variables is 30 years, as defined by the World Meteorological Organization. The relevant quantities are most often surface variables such as temperature, precipitation and wind. Climate in a wider sense is the state, including a statistical description, of the climate system.*

While the above definitions are community property, for completeness we also quote in Appendix B definitions from individuals, taken from celebrated books.

A useful observation is that all definitions use the term "average". Thus, by its definition, climate is a statistical concept.

By scrutinizing the definitions, several questions may arise. A first is: Why "*at least a 30-year period*"? Is there anything special about 30 years? It appears that this reflects a historical belief that 30 years are enough to smooth out random weather components and establish a constant mean. In turn, this reflects a perception of a constant climate, and a hope that 30 years would be enough for a climatic quantity to stabilize to a constant value. It can be conjectured that the number 30 stems from the central limit theorem and in particular the common (but not quite right) belief that the sampling distribution of the mean is normal for sample sizes over 30 (e.g., [23]). Such a perception roughly harmonizes with classical statistics of independent events. This perception is further reflected in the term anomaly (from the Greek $\alpha\nu\omega\mu\alpha\lambda\acute{\iota}\alpha$, meaning abnormality), commonly used in climatology to express the difference from the mean. Thus, the dominant idea is that a constant climate would be the norm, and a deviation from the norm would be an abnormality, perhaps caused by an external agent. However, such belief is incorrect and inconsistent with the reality of an ever-changing climate. This was pointed out almost 50 years ago by Lamb [24]:

*the view, regarded as scientific, which was widely taught in the earlier part of this century, that climate was essentially constant apart from random fluctuations from year to year was at variance with the attitudes and experience of most earlier generations. It has also had to be abandoned in face of the significant changes in many parts of the world that occurred between 1900 and 1950 and other changes since.*

Clearly, however, even the later generations were not able to get rid of this "view regarded as scientific", which remains dominant.

A second question inspired by Climate Prediction Center's definition (point (2) above) is: Why the climate taken over 30 or 1000 years is different? The obvious reply is: Because different 30-year periods have different climate. This contradicts the tacit belief of constancy and harmonizes with the perception of an ever-changing climate. With the latter perception, Herbertson's idea that "*climate is what we expect, weather is what we get*" can be reformulated as "*weather is what we get immediately, climate is what we get if we keep expecting for a long time*" [25].

As many of the above definitions refer to weather, it is useful to clarify its meaning, noting that it represents a popular notion, often used with respect to its effects upon life and human activities, rather than a rigorous scientific one. Interestingly, in its colloquial use in Greek and Romance (Neo-Latin) languages, weather is almost indistinguishable from time (Greek: καιρός; Italian: tempo; French: temps, météo; Spanish: tiempo, clima; Portuguese: tempo, clima). On the other hand, in English and Greek, weather refers to short-scale variations in the atmosphere and is distinguished from climate; note however that in colloquial Spanish and Portuguese there is no such distinction. In scientific terms, the definition given by the WMO [21] is this:

> *W0410 weather–State of the atmosphere at a particular time, as defined by the various meteorological elements.*

## 4. Toward a Rigorous Definition of Climate

The importance of definitions is highlighted by the quotation by Luzin given in the beginning of the paper. However, this importance may not have been widely appreciated as exemplified by the popularity (almost four thousand citations in Google Scholar) of the paper entitled "Stationarity is dead" [26], which does not refer to a definition of stationarity at all. Even worse, the use of the term "stationarity" in this paper is not consistent with its existing scientific definition, as thoroughly explained by Koutsoyiannis and Montanari [27,28]. A second example of disdaining definitions is Mandelbrot's [29] opinion that absence of a definition "ought not create concern and steal time from useful work".

Yet one may wish to adhere to the principle that definitions are a necessary element of the scientific method. In this case, one may wish to revisit the definition of climate, given the problems already examined in Section 3 and Appendix B. In this respect, Koutsoyiannis [30] attempted to give a definition of climate in a hierarchical manner (avoiding circular logic) starting from the concept of climatic system, as follows:

*Climatic system* is the system consisting of the atmosphere, the hydrosphere (including its solid phase—the cryosphere), the lithosphere and the biosphere, which mutually interact and respond to external influences (system inputs) and particularly those determining the solar radiation reaching the Earth, such as the solar activity, the Earth's motion and the volcanic activity.

*Climatic processes* are the physical, chemical and biological processes, which are produced by the interactions and responses of the climatic system components through flows of energy and mass, and chemical and biological reactions.

*Climate* is a collection of climatic processes at a specified area, stochastically characterized for a range of time scales.

It is stressed that the stochastic characterization, appearing in the proposed definition, does not refuse the existence of physical dynamical laws and causal relationships among the elements of climate, nor does it equate climate with dice. It is most surprising that one and a half centuries after the explanation of entropy within the probability theory, and the establishment of statistical thermophysics, many still confuse stochastics with pure randomness, and contrast physics to stochastics or statistics.

The notion of stochastic characterization collectively encompasses all related concepts of the scientific areas of probability, statistics and stochastic processes. Since climate is not static but dynamic (and by now the evidence that climate has ever been changing is overwhelming), it is better to think of it as a stochastic concept. In stochastics change, and hence time, have a hypostasis that is typically absent in statistics. The direct analogy is dynamics vs. statics.

Naturally, the stochastic characterization includes the statistics used in other definitions, such as averages, variability, extremes, etc. However, there is a big difference in our definition from standard definitions. By not distinguishing whether "average" or "mean" refer to the true mean or the temporal mean, the latter definitions are affected by ambiguity. The common interpretation is that they refer to the true (or ensemble) mean of the probabilistic vocabulary, in which case the mean is a number, a constant.

However, in our definition climate is represented as a time average, is dependent on the time scale of averaging and keeps depending on time per se, as also happens with weather. A time average of a stochastic process (originally defined in continuous or discrete time) is not a number but a stochastic process per se. Thus, both instantaneous processes (or discrete time processes at time-scales pertinent to weather) and climatic processes are stochastic processes distinguished only by the time scale of discretization. If the time-scale is small (e.g., hourly or daily) or tends to zero (instantaneous process) then we speak about weather, and this harmonizes with the definition by the WMO [21], which is kept unchanged here. If the time-scale is large, greater than the annual, then we have climate. In this respect climate is a macroscopization of weather by removing the details through time averaging.

We recall that a stochastic process $\underline{x}(t)$ at continuous time $t$ is a family of stochastic (random) variables $\underline{x}$ indexed by time $t$. (Note that underlined symbols denote stochastic variables or stochastic processes.) Formally, it is defined by means of its $n$th order distribution function for an arbitrary natural number $n$:

$$F(x_1, x_2, \ldots, x_n; t_1, t_2, \ldots, t_n) := P\{\underline{x}(t_1) \le x_1, \underline{x}(t_2) \le x_2, \ldots, \underline{x}(t_n) \le x_n\} \quad (1)$$

In application, since a process is never observed or simulated in continuous time but in discrete times, usually assumed to be equidistant with temporal resolution $D$, i.e., $t_\tau = \tau D$, for an integer $\tau$, we use a discrete time representation of the process, i.e.,

$$\underline{x}_\tau := \frac{1}{D} \int_{(\tau-1)D}^{\tau D} \underline{x}(t) \mathrm{d}t \quad (2)$$

We further define the cumulative process of $\underline{x}(t)$ or $\underline{x}_\tau$, for discrete time scale $\kappa$ or a continuous one $k := \kappa D$, as:

$$\underline{X}(k) \equiv \underline{X}_\kappa := \underline{x}_1 + \underline{x}_2 + \ldots + \underline{x}_\kappa = \int_0^{\kappa D} \underline{x}(t) \, \mathrm{d}t \quad (3)$$

The time average of the original process $\underline{x}_\tau$ for discrete time scale $\kappa$ is

$$\underline{x}_\tau^{(\kappa)} := \frac{\underline{x}_{(\tau-1)\kappa+1} + \underline{x}_{(\tau-1)\kappa+2} + \ldots + \underline{x}_{\tau\kappa}}{\kappa} = \frac{\underline{X}_{\tau\kappa} - \underline{X}_{(\tau-1)\kappa}}{\kappa} \quad (4)$$

We also recall that a stochastic process is a model, an abstract mathematical construction, a family of stochastic variables (behaving like functions or sets) rather than a sequence of numbers. However, a realization $x_\tau$ of a stochastic process $\underline{x}_\tau$ is a sequence of numbers and is called a time-series. A time-series can be formed either by observation of the process being modelled or synthesized by implementing the model, i.e., the stochastic process. In the former case we can only have a single time series as Nature does not repeat herself. In the latter case, we can have an ensemble of time-series with as many members as we wish, obtained by repetitive model runs.

The mathematical operations of the above equations can also be performed on the time-series $x_\tau$ instead of the stochastic process $\underline{x}_\tau$, and in this case the results (e.g., $x_\tau^{(\kappa)}$) are numbers. These can be used to illustrate the concepts we discuss, e.g., by plotting $x_\tau^{(\kappa)}$ vs. $\tau$. Note that the stochastic process $\underline{x}_\tau^{(\kappa)}$ cannot be plotted because it is not composed of numbers.

For our illustration, we choose the process of precipitation because (a) together with temperature, it is one of the two key processes used for climate classification and (b) its variability is much higher than temperature and its behaviour on extremes much more interesting. Furthermore, as a location for our illustration we choose Bologna, Italy (44.50° N, 11.35° E, 53.0 m) because it has one of the longest daily records of rainfall and

temperature worldwide and thus enables insights into the evolution of climate. The time-series of observations is available online in the frame of the Global Historical Climatology Network–Daily (GHCN-D, [31]). It is uninterrupted for the period 1813–2007, i.e., 195 years. For the most recent period, 2008–2018, daily data are provided by another online data repository [32], and thus the record length increases to 206 years in total.

When $k := \kappa D$ is small, the process $\underline{x}_\tau^{(\kappa)}$ represents weather. Illustration of the evolution of weather at the hourly and daily scales is given in Figure 3. Conversely, when $k$ is large, $\underline{x}_\tau^{(\kappa)}$ represents climate, as illustrated in Figure 4 (upper panel), for $k = 10$ and 30 years.

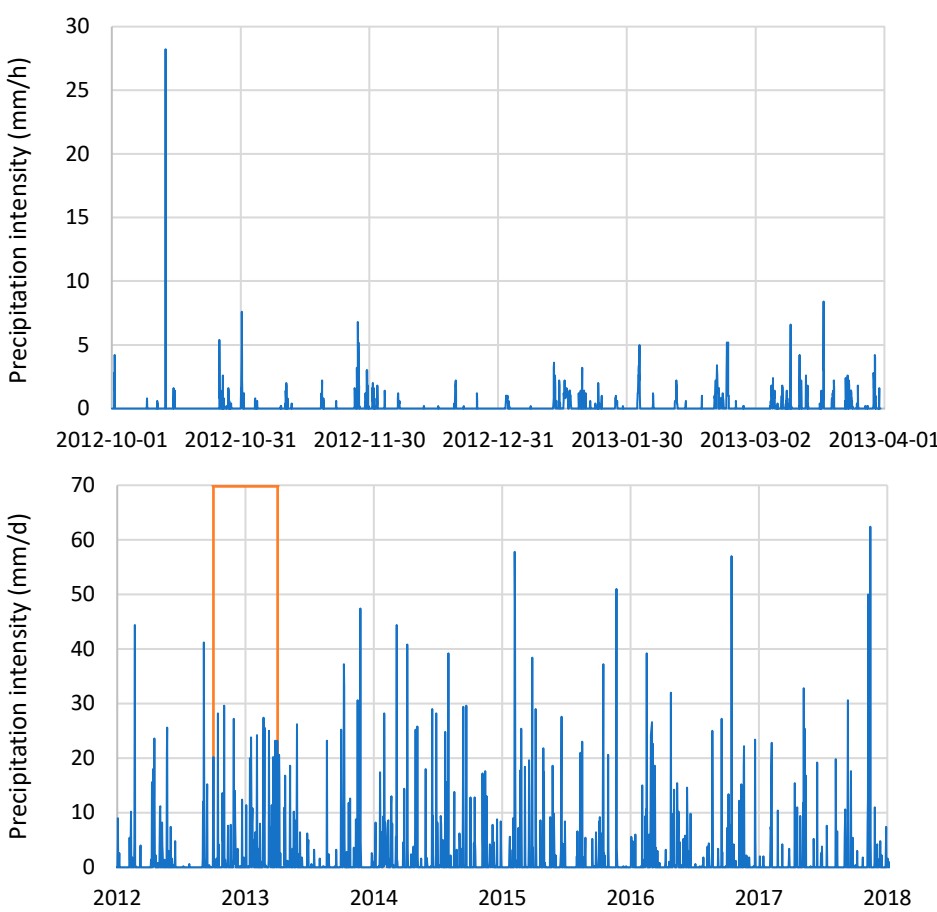

**Figure 3.** Evolution of precipitation in Bologna, as a weather element, seen at (**upper**) hourly time scale for six months and (**lower**) daily time scale for six years (the orange rectangle is the time domain of the upper graph).

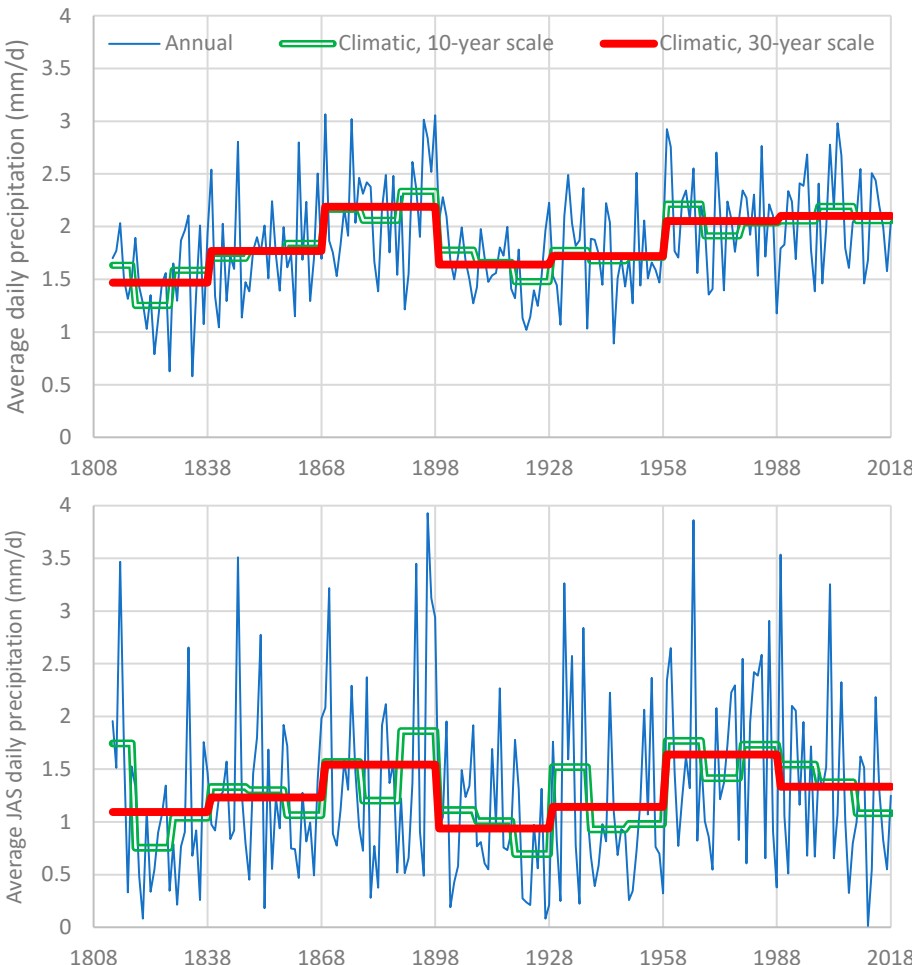

**Figure 4.** (**upper**) Evolution of average daily precipitation in Bologna, as a climatic element, seen at the annual and the climatic time scales of 10 and 30 years; (**lower**) as in upper panel but for a time window of the three summer months, JAS.

Apparently, as seen in Figure 4, the variability decreases as the time-scale increases, but it never becomes zero. This is the case even for time scales of millions of years [33]. The variability is quantified by the variance of the process:

$$\gamma_\kappa := \text{var}\left[\underline{x}_\tau^{(\kappa)}\right] \tag{5}$$

Clearly, this is a function of the time-scale $\kappa$ which is termed the climacogram of the process, from the Greek climax (κλίμαξ, meaning scale) [34].

For sufficiently large $\kappa$ (theoretically as $\kappa \to \infty$), we may approximate the climacogram as:

$$\gamma_\kappa \propto \kappa^{2H-2} \tag{6}$$

where $H$ is termed the Hurst parameter. The theoretical validity of such (power-type) behaviour of a process was implied by Kolmogorov (1940 [35,36]). The quantity $2H–2$ is visualized as the slope of the double logarithmic plot of the climacogram for large time-scales. In a random process, $H = 1/2$, while in most natural processes $1/2 \leq H \leq 1$, as first observed by Hurst in 1951 [37]. This natural behaviour is known as (long-term) persistence or Hurst-Kolmogorov (HK) dynamics. A high value of $H$ (approaching 1) indicates enhanced presence of patterns, enhanced change and enhanced uncertainty (e.g., in future predictions). A low value of $H$ (approaching 0) indicates enhanced fluctuation or antipersistence.

The climacogram of the precipitation in Bologna is depicted in Figure 5, which reflects the variability on a continuum of time-scales referring to both weather and climate.

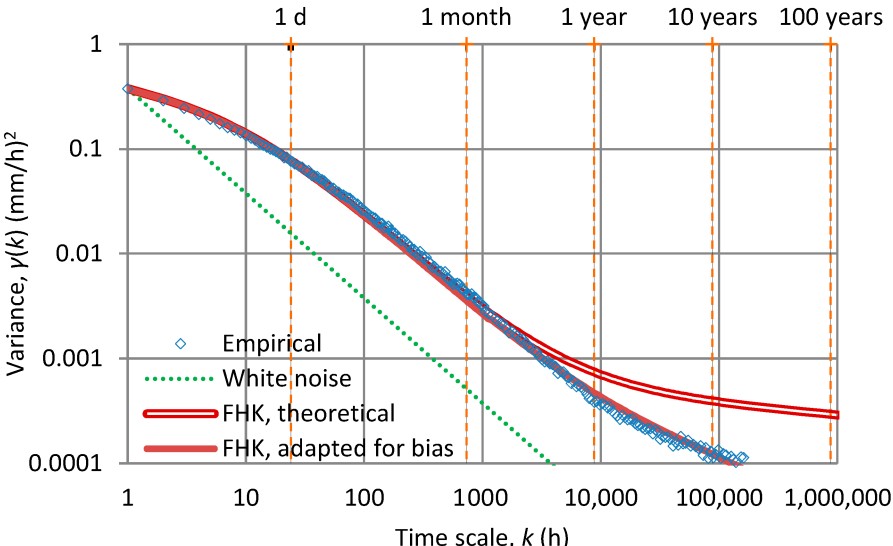

**Figure 5.** Climacogram of precipitation in Bologna; both hourly and daily data have been used to estimate the empirical climacogram. FHK stands for filtered Hurst-Kolmogorov process and has the mathematical expression of Equation (7) with fitted parameters $H = 0.95$, $M = 0.05$, $\alpha = 16.4$ h, $\lambda_1 = 0.000864$ mm$^2$/h$^2$ and $\lambda_2 = 1.506$ mm$^2$/h$^2$.

The theoretical model fitted to the empirical climacogram of Figure 5, termed the filtered Hurst-Kolmogorov process of Cauchy-Dagum type (FHK-CD; [30,38]), has the mathematical expression:

$$\gamma(k) = \lambda_1 \left(1 + \frac{k}{\alpha}\right)^{2H-2} + \lambda_2 \left(1 - \left(1 + \frac{\alpha}{k}\right)^{-2M}\right) \qquad (7)$$

In addition to the Hurst parameter $H$, which characterizes the global scaling behaviour, when $k \to \infty$ (pertinent to the climate), the model includes a second scaling exponent $M$ characterizing the local scaling or smoothness or fractal behaviour when $k \to 0$ (pertinent to the weather). Furthermore, the model includes a time scale parameter $\alpha$ and two state scale parameters $\lambda_1$ and $\lambda_2$. The large Hurst parameter estimated ($H = 0.95$) implies large estimation bias at large climatic scales, which was considered in the fitting of the model and shown in Figure 5.

The climacogram of the simple and familiar case, in which the process is white noise (purely random), is also plotted in Figure 5. This would imply rapid increase of climatic variability and would be consistent with the idea of a stable climate. It can be conjectured that it is the white noise paradigm that misled climatologists to form the idea of a stable climate that needs an external agent to change it. However, the real climate is not white noise. As seen in Figure 5 the white noise model entails a reduction of variability of three orders of magnitude when we move from the hourly time scale to that of a couple of months. However, in the real climate we need a 100-year climatic scale to achieve this reduction.

As geophysical processes are affected by the double periodicity, daily and annual, related to the Earth's motion, it is a common requirement in several applications to study a specified window of time for each year (e.g., one or more adjacent months), or even a specific part of the day (e.g., some morning hours in a certain month). It is easy, then, to specify an indication function $I(t)$, taking the value 1 whenever $t$ belongs to the specified time window, and 0 otherwise. Assuming that a temporal resolution of $D = 1$ year and

denoting $D_{\mathrm{w}} := \int_0^D I(t)\mathrm{d}t$ the length of the window, we define the windowed process at the annual time scale as:

$$\underline{w}_\tau := \frac{1}{D_{\mathrm{w}}} \int_{(\tau-1)D}^{\tau D} \underline{x}(t)I(t)\mathrm{d}t \tag{8}$$

We can then define the discrete time process at any climatic time scale $\kappa > 1$, i.e., $\underline{w}_\tau^{(\kappa)}$, by applying (4). An illustration of this idea is provided in the lower panel of Figure 4 for the precipitation in Bologna during the summer months of July-September.

Furthermore, when we are interested about extremes of a certain process, we can replace the time averaging operation with taking the maximum or minimum, i.e.,:

$$\underline{y}_\tau^{(\kappa)} := \max\left(\underline{x}_{(\tau-1)\kappa+1}, \underline{x}_{(\tau-1)\kappa+2}, \dots, \underline{x}_{\tau\kappa}\right), \underline{z}_\tau^{(\kappa)} := \min\left(\underline{x}_{(\tau-1)\kappa+1}, \underline{x}_{(\tau-1)\kappa+2}, \dots, \underline{x}_{\tau\kappa}\right) \tag{9}$$

Assuming that the time scale is greater than annual, the stochastic processes $\underline{y}_\tau^{(\kappa)}$ and $\underline{z}_\tau^{(\kappa)}$ are climatic. Illustration of $\underline{y}_\tau^{(\kappa)}$ for the precipitation in Bologna is given in Figure 6. This can also be combined with windowing on a specific season as also shown in the same figure.

To define another interesting case of a climatic stochastic process, let us consider a specific probability referring to the marginal distribution of $\underline{x}_\tau$ (or $\underline{w}_\tau$ if we consider a subperiod of the year), such as $\Pi := P\{\underline{x}_\tau \leq c\}$, where $c$ is a constant. This $\Pi$ is a regular variable taking on a particular value, rather than a stochastic variable. However, if we estimate it from a particular time-series, which is a realization of $\underline{x}_\tau$, then the estimates change for different $\tau$, so that they can be represented as a family of stochastic variables $\underline{\pi}_\tau$, or else a stochastic process. If then we average this process at any climatic time scale $\kappa > 1$, i.e., $\underline{\pi}_\tau^{(\kappa)}$, by applying (4), then we get a climatic stochastic process.

For the rainfall process, if we set the threshold $c = 0$, we obtain the probability dry, which provides a way to study the low extremes. (Note that the process $\underline{z}_\tau^{(\kappa)}$ in precipitation is zero by identity.) Illustration is provided in Figure 7 for the probability dry in Bologna.

In all climatic processes illustrated, the climatic variability is omnipresent. Further illustration of that variability is provided for the 30-year climatic scale in Table 2 for all processes that are used for the Köppen climate classification, including those referring to temperature, whose time-series of observations are equally long as in precipitation. We may observe that the 30-year climatic value of annual precipitation varied from 536.4 mm (1813–1838) to 798.7 mm (1869–1898), a ratio of 1:1.5. No monotonic trend appears in the climatic values of precipitation. In temperature there also appear fluctuations, but with a warming trend in the most recent years. Interestingly however, this warming is mostly a result of increasing of the lowest temperatures (milder winters; see also Appendix C).

Despite the significant climatic fluctuations, the Köppen climate type of Bologna remains the same through all seven 30-year climatic periods, i.e., Cfa: temperate without a dry season and with hot summer.

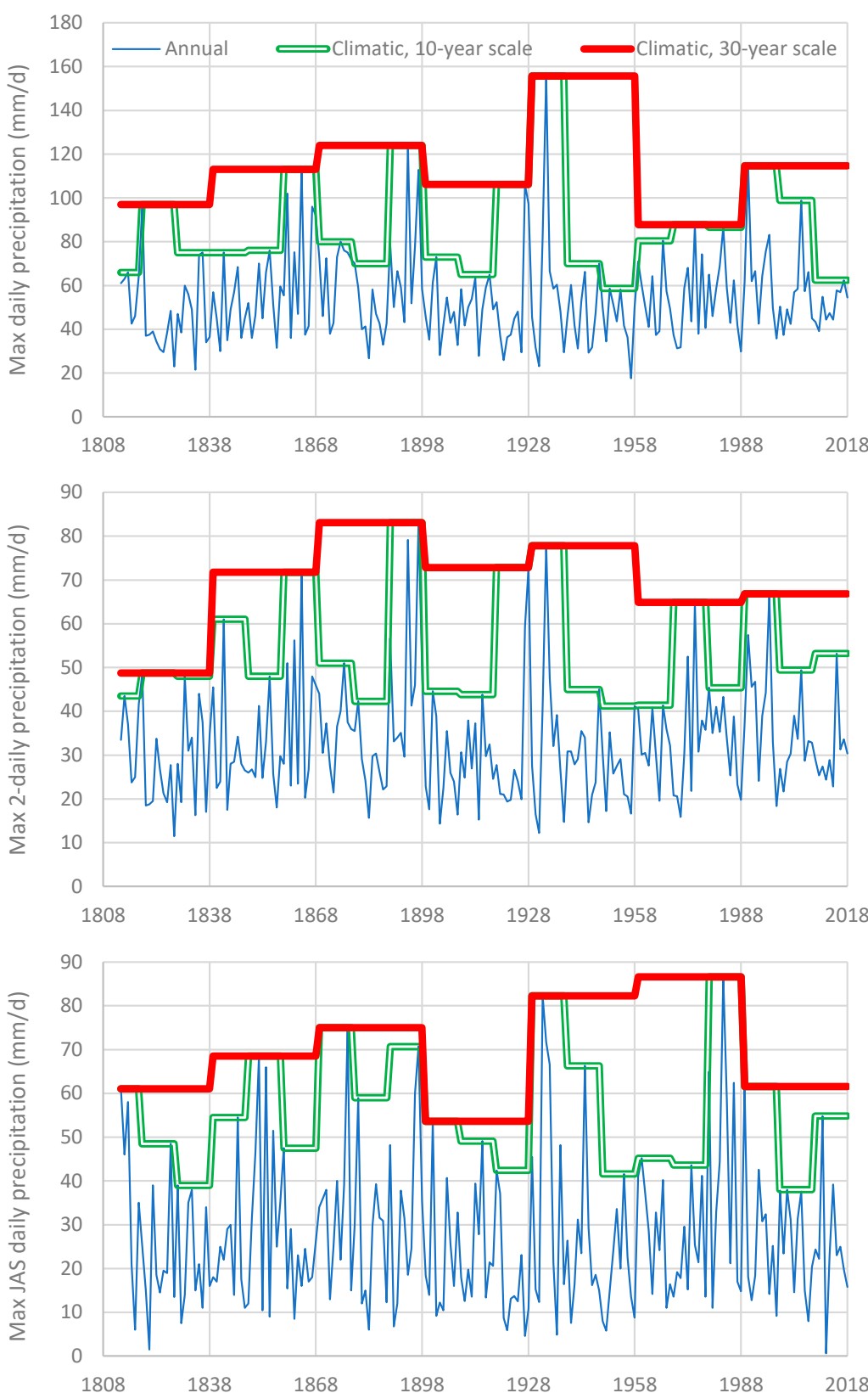

**Figure 6.** (**upper**) Evolution of the maximum daily precipitation in Bologna, as a climatic element, seen at the annual and the climatic time scales of 10 and 30 years; (**middle**) as in the upper panel but for the maximum two-daily precipitation; (**lower**) as in upper panel but for a time window of the three summer months, July, August, September (JAS).

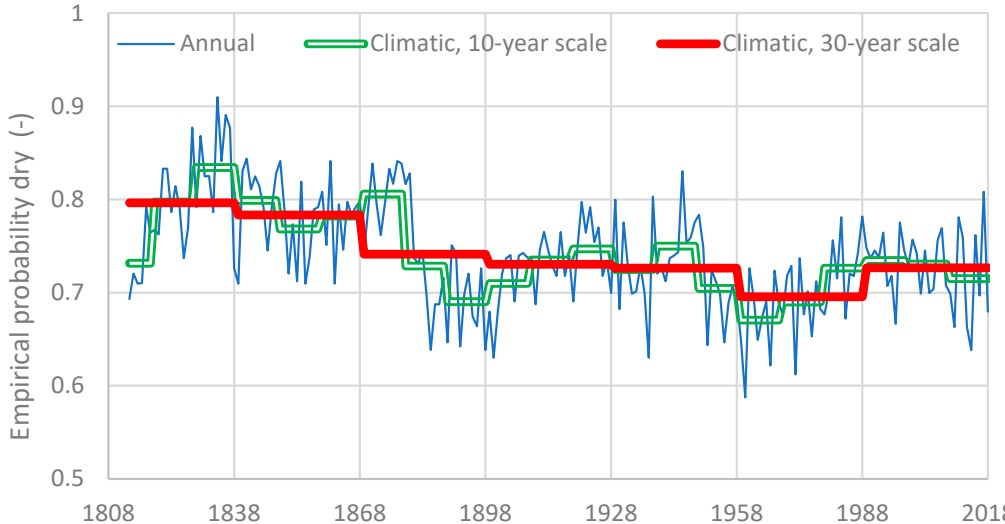

**Figure 7.** Evolution of the empirical probability dry of daily precipitation (relative frequency of days without precipitation) in Bologna, as a climatic element, seen at the annual and the climatic time scales of 10 and 30 years.

Because in Bologna there is no dry season, the effect of seasonality on precipitation is low and can be neglected in the stochastic characterisation of the precipitation process. The negligible effect is manifest in the climacogram of Figure 5, where a monotonic decrease is observed for the entire range of scales. However, in other climatic zones the seasonality is prominent. Even in Bologna, in the temperature process, which has been used for the climatic classification in Table 2, the effect of seasonality is substantial. The climacogram can no longer be fully described by Equation (7).

**Table 2.** Climatic time series of temperature and precipitation in Bologna that are used for the Köppen climate classification.

| Statistic | 1813–1838 | 1839–1868 | 1869–1898 | 1899–1928 | 1929–1958 | 1959–1988 | 1989–2018 |
|---|---|---|---|---|---|---|---|
| Temperature, annual (°C) | 13.9 | 13.4 | 13.7 | 13.8 | 14.0 | 13.8 | 15.3 |
| Temperature, hot half-year (°C) * | 20.9 | 20.4 | 20.5 | 20.3 | 21.1 | 20.2 | 21.8 |
| Temperature, cold half-year (°C) | 6.8 | 6.3 | 6.8 | 7.3 | 7.0 | 7.3 | 8.7 |
| Temperature, hottest month (°C) | 25.4 | 24.8 | 25.3 | 24.6 | 25.5 | 24.5 | 26.2 |
| Temperature, coldest month (°C) | 1.4 | 1.3 | 1.7 | 2.8 | 2.0 | 2.6 | 4.4 |
| Number of months with temperature >10 °C | 7 | 7 | 7 | 7 | 7 | 7 | 8 |
| Precipitation, annual (mm) | 536.4 | 646.4 | 798.7 | 598.4 | 628.0 | 750.1 | 767.2 |
| Precipitation, hot half-year (mm) | 265.7 | 318.4 | 378.8 | 281.8 | 276.1 | 351.3 | 365.5 |
| Precipitation, cold half-year (mm) | 270.7 | 328.0 | 420.0 | 316.7 | 351.8 | 398.8 | 401.7 |
| Precipitation, driest month of hot half-year * (mm) | 30.2 | 34.0 | 40.9 | 26.0 | 32.9 | 46.5 | 40.1 |
| Precipitation, wettest month of hot half-year (mm) | 62.4 | 74.1 | 84.3 | 62.5 | 66.1 | 65.2 | 76.6 |
| Precipitation, driest month of cold half-year * (mm) | 30.1 | 27.9 | 40.0 | 30.9 | 44.6 | 47.3 | 40.2 |
| Precipitation, wettest month of cold half-year (mm) | 60.7 | 87.4 | 114.2 | 74.9 | 90.3 | 86.7 | 88.4 |
| Köppen climate type | Cfa | Cfa | Cfa | Cfa | Cfa | Cfa | Cfa |

* The hot half-year is defined as the six-month period of AMJJAS (April, May, June, July, August, September) and the cold half-year as that of ONDJFM (October, November, December, January, February, March).

However, by adding a harmonic oscillation described as $x(t) = \sqrt{2\lambda} \cos(2\pi(t+b)/T)$, where $T$ is the period and $b$ is the phase, $0 \leq b \leq T$, we can achieve a satisfactory characteri-

zation of seasonality. The resulting climacogram of the harmonic oscillation, which should be added to that of Equation (7), is [38]:

$$\gamma_T(k) = \begin{cases} \frac{\lambda_T T^2}{\pi^2 k^2} \sin^2\left(\frac{\pi k}{T}\right), & k \neq (m + \frac{1}{2})T \\ \frac{2\lambda_T T^2}{\pi^2 k^2} \sin^2\left(\frac{2\pi b}{T}\right), & k = (m + \frac{1}{2})T \end{cases} \tag{10}$$

where $m \in N_0$. Clearly, for increasing time-scale $k$ the contribution of $\gamma_T(k)$ vanishes.

The period $T$ is one year. Depending on the problem, additional harmonics with smaller periods may be required for a better representation, while for a sufficient representation at subdaily scales at least one more harmonic with period $T = 1$ d needs to be added.

The effect and the modelling of seasonality are shown in Appendix C using, for illustration, different variables, the maximum and minimum daily temperature, and, as study cases, different sites, Vienna (Austria) and Melbourne (Australia), which again are among those with the longest time series for these particular variables.

## 5. Climate and Water

As evident in its definition, the climatic system is subject to external influences (system inputs) and particularly those determining the solar radiation reaching the Earth, such as the solar activity, the Earth's motion and the volcanic activity. Changes in the solar irradiance (which is reflected in the sunspot number and is maximum and minimum when the sunspot number is maximum and minimum, respectively), as well as in the solar and terrestrial magnetic fields, are known to influence climate [39,40]. It has been suggested that even the galactic cosmic ray flux may be a climate driver via solar wind modulation [41,42]. The oscillations of the Earth's orbit, namely variations in eccentricity, axial tilt, and precession (with the latter having been discovered by Hipparchus, as already mentioned in Section 2) are important drivers of climate and are collectively known as the Milanković cycles, after the Serbian civil engineer Milutin Milanković (1879–1958) who studied them [43–45]. Recently, it has been demonstrated in a persuasive manner by Roe [46] that it is the effect of the Milanković cycles, rather than of atmospheric $CO_2$ concentration, that explains the large scale climatic evolution (namely, the glaciation process) in the Quaternary.

These external drivers have changed substantially through the lifetime of Earth. According to Kuhn et al. [47], four billion years ago the solar irradiance was about 80% of the current value (or, according to other estimates, 75%), the Earth's rotation rate was 170% of the current, the land area was very small, less than 4% of the current value, and the atmospheric $CO_2$ concentration was about three orders of magnitude higher or more (up to 250,000%; see this information in a combined graph in [48]). Note that even in the Cenozoic (the last 65 million years) the atmospheric $CO_2$ concentration varied by more than two orders of magnitude (see graph in [49]). Amazingly however, despite these cosmogonic changes, the temperature remained fairly constant (varying by only 10%, which is equivalent to 29 K) during all these four billion years. For example, evidence shows the existence of liquid water on Earth even in the earliest period, when the solar activity was smaller by 20–25%, a puzzle known as the faint young Sun problem [50]. One may attribute the temperature stability to the regulating properties and processes of the climatic system, and may conjecture that the hydrosphere in particular must have played some important role in it.

As stated in the WMO's definition of climate quoted above, the typical use of the term climate refers to the atmosphere only, leaving out the hydrosphere and the other parts of the climatic system. However, since the climatic system includes the hydrosphere, there is no reason to exclude the hydrological processes from the climatic processes. Therefore, our definition includes them. Nevertheless, to give more emphasis on the inclusion of hydrological processes, the term hydroclimatic is often used, which gives additional clarity, but is rather a redundancy as the hydrosphere is already included in the climate system.

The established idea is that the hydrology of an area, including the water balance and hydrological extremes, is affected unidirectionally by climate. This idea is further expanded to establish a linear causality chain of the type: human $CO_2$ emissions $\rightarrow$ increasing concentration of atmospheric $CO_2$ $\rightarrow$ increasing temperature $\rightarrow$ changes in hydrological processes and water balance. This is evident in the popular practice of studying the so-called climate change impacts on hydrological processes. However, this is a naïve idea that does not correspond to physical reality.

The importance of the effect of the different greenhouse gases, or other agents affecting climate, is not necessarily related to the research efforts and scientific publications on each of them. Arguably, the fact that the $CO_2$ has been so heavily and repeatedly studied, particularly in paleoclimatology studies (e.g., [49,51–57]), does not suggest that it is more important a greenhouse gas than water. A simpler explanation is that $CO_2$ concentration is easier to study because its change in time is smoother, because its spatial heterogeneity is much lower than that of water vapour, and because it can be detected in ice and sediment cores, stomatal complexes, etc.

Here we argue that water is the most crucial element determining climate (e.g., [58,59]), or as put by Poyet [60], "Water is the main player". We list epigrammatically some of the reasons justifying it:

- *Abundance*. The water in the oceans amounts to $1.34 \times 10^9$ Gt [61] while additional quantities are stored in the soil, ground and glaciers (which generally are not in turbulent motion, see below) and much smaller quantities of liquid water are on land. For comparison the mass of air in the atmosphere is $5.14 \times 10^6$ Gt (of which 12,500 Gt is water vapour) [62], i.e., 260 times smaller than the mass of water in turbulent motion.

- *Heat storage*. Water's specific heat (or heat capacity) is 4218 and 2106 J kg$^{-1}$ K$^{-1}$ for the liquid and solid phase, respectively, and 1463 and 1924 J kg$^{-1}$ K$^{-1}$ for the gaseous phase for constant volume and constant pressure, respectively [63]. These figures are considerably larger than the specific heat of dry air (707 and 1004 J kg$^{-1}$ K$^{-1}$ for the gaseous phase for constant volume and constant pressure, respectively), as well as of the dry soil (typically 800 J kg$^{-1}$ K$^{-1}$). As a result of its high heat capacity and abundance, water is the element that determines the heat storage and hence the climate of the Earth. For example, in the last forty years, accumulated heat in the oceans is 94% of the total, with the remaining 6% distributed to ice, land and atmosphere (see details in Appendix D). For the same reason, water has been called the climatic thermostat of the Earth [64].

- *Heat exchange*. For the average temperature of Earth, the specific latent heat of water evaporation (calculated from "Equation (40)" in [65]) is 2.47 MJ/kg, much higher than in other common substances. Also, the specific latent heat of ice melt (solid water fusion) is 0.334 MJ/kg, again higher than in other common substances. Combined with the fact that water abounds on Earth in all three phases, these high values of phase change energy make water the thermodynamic regulator of climate. In particular, heat exchange by evaporation (and hence the latent heat transfer from the Earth's surface to the atmosphere) is the Earth's natural locomotive, with the total energy involved in the hydrological cycle being 1290 ZJ/year, corresponding to an energy flux density of 80 W/m$^2$ [59]. Compared to human energy production (0.612 ZJ/year for 2014), the total energy of the natural locomotive is 2100 times higher than that of the human locomotive [59].

- *Shortwave radiation regulation*. An interesting property of water is the spectacular differences, among the different phases and formations, in the reflecting properties of the incoming sunlight, that is, the albedo. Thus, the albedo of liquid water is only 5% near the equator (but increases up to 76% near the poles for winter months [66]), while for ice it is 32–38%, and increases further to 45% for old snow and up to 85% for fresh snow. Even though water vapour is transparent to shortwave radiation, the water has a huge effect on the radiation properties of the atmosphere through the clouds. The albedo of clouds varies by even larger ranges, from 10% to more than

80% [67], depending on drop sizes, liquid water or ice content, thickness of the cloud, and sunray's angle. The importance of the albedo can be understood by this example: The incoming solar radiation is 341 W/m$^2$ [68]; this is the solar constant divided by four, in order to take the average over the entire Earth's surface of the amount of sunlight power that reaches the atmosphere. Given that, a tiny change of 1‰ in the albedo due to differences in the presence of water formations (snow, ice, clouds) results in an "imbalance" of 0.34 W/m$^2$. This is of the same order or magnitude of the average imbalance (net absorbed energy) of the Earth in the last 50 years, allegedly attributed to the increase of the $CO_2$ concentration (see calculations in Appendix D). We note that the 10-year climatic variability of albedo in the last 40 years was 1% (see calculations in Appendix E), i.e., 10 times higher than in our hypothetical change of 1‰ of our example.

- *Longwave radiation regulation.* Even though in the common perception it is carbon dioxide ($CO_2$) that determines the greenhouse effect of the Earth, recent studies (Schmidt et al. [69]) attribute only 19% of the longwave radiation absorption to $CO_2$ against 75% of water vapour and clouds, or a ratio of 1:4. According to other estimations, the importance of $CO_2$ as a greenhouse gas is even lower and that of water higher [60]. Another misconception, common in nonexperts, is that atmospheric $CO_2$ is the product of human emissions, while in fact the latter contribute only 3.8% to the global carbon cycle [48]. Water, as a universal solvent, plays some role for the presence of $CO_2$ in the hydrosphere and the atmosphere. This role is described by Henry's law [70], according to which increase of water temperature results in decreasing solubility of $CO_2$ in water and hence $CO_2$ degassing from the ocean. To this physical process we should add the fact that increase of temperature ($T$) results in increased respiration of the biosphere on land and sea. Based on such considerations, and using reliable instrumental measurements of global $T$ and $CO_2$ concentration covering the time interval 1980–2019, a recent study [49] found that in the relationship of $CO_2$ and temperature, the dominant causality direction is $T \rightarrow CO_2$, rather than the other way round, despite the latter being the common perception.

- *Turbulent motion.* The climate is generated by the everlasting turbulent motion of two fluids, water and air. The turbulent dynamics in the circulation of both fluids is much more complex and less well known than thermodynamics [71]. The motion of both fluids is thus inherently uncertain and produces patterns that can hardly be predicted in advance. In the atmosphere, the complexity in motion is further perplexed due to the presence of water in the air, in the form of vapour and clouds, which play a crucial role. For example, recent studies [72,73] point to the role of water vapour and clouds in warming the Arctic environment. Because humans (as contrasted to fish) live in contact with the atmosphere, the motion in the atmosphere is better observed and studied than in the hydrosphere. This does not mean that the motion in the latter, particularly the large-scale fluctuations, is less important for climate. For example, the rhythm of coupled ocean–atmosphere fluctuations, such as the El Niño–Southern Oscillation (ENSO), Atlantic Multidecadal Oscillation (AMO) and Interdecadal Pacific Oscillation (IPO), significantly influences the variability of global mean annual temperature [74].

- *Elixir of life.* The phrase "water is the elixir of life" appeared in the 19th century in a book by Allen [75], who attributes it to "Scriptures" and asserts that three fourths of diseases are caused by abuse of water. More recently, Eagleson [76] used the same phrase justifying it by the fact that water is a universal solvent and that cell membranes are permeable only to dissolved substances. Thus, the biosphere depends on water, whose presence determines the type and extent of ecosystems. In turn, the ecosystems affect climate at large through the carbon and oxygen cycles (where the vast majority of the $CO_2$ and $O_2$ emissions are products of life, through respiration and photosynthesis, respectively), and their contribution in the water cycle (transpiration) and in the energy cycle (photosynthesis). Humans, as part of the biosphere, also interact with water and

climate; they are affected by them and affect them. Since the invention of technology in the Neolithic age and, in particular, the establishment of perennial agriculture and the advent of urbanization, these human effects became larger in terms of land-use change and contribution in the mass and energy cycles. Moreover, after the industrial revolution, the anthropogenic effects are marked and in certain aspects unsustainable. Notably, human interventions on land and on water bodies may have much more substantial effects on the entire Earth than the infamous fossil fuel burning and the resulting $CO_2$ emissions. For example, considering sea level rise, the most prominent anthropogenic signal is the increased (and unsustainable) exploitation of groundwater, which transfers to the sea huge masses of water earlier stored in the land [59]. To describe the growing impacts of human activities on Earth, including in geology and ecology, P. Crutzen and E. Stoermer proposed the term *anthropocene* for the current geological epoch [77]. However, the proposal has not been ratified by the International Commission on Stratigraphy, nor by the International Union of Geological Sciences (note that neither of the proposers was a geologist; Crutzen was an atmospheric chemist and Stoermer a biologist). Nonetheless, the term is quite popular in other disciplines in an environmental, bioethical and political context [78], with the latter sometimes related to "urging to action". The term was criticised by M. Sagoff [79], who asserted that the underlying idea that humans rule Nature "accomplishes a counter-Copernican revolution", in which "The Anthropocene makes humanity great again", hence implying that the term is equivalent to "Narcisscene" (which he uses in his article title).

Putting aside socio-political issues reflected in the above contentions, we may return to the scientific discussion of the relationship of climate and water, and atmosphere and hydrosphere. Certainly, it makes sense to view atmospheric phenomena as the cause of hydrological ones when the time-scale is small (pertinent to weather). However, the above discourse supports a conclusion that the opposite is the case when the time-scale is large (pertinent to climate). Hopefully, this could be assimilated by hydrologists who in climate studies have currently delimited their own role to "climate impactologists", taking as input the climate model outputs and producing results for the future of water on Earth as if the inputs were credible. However, as a result of the "cart before the horse" approach, which confuses or reverses roles and causality directions, those inputs are not credible. As shown in a series of publications, the climate model outputs are irrelevant to reality and thus not hydrologically useful for all time-scales, from sub-annual to climatic, and for a variety of spatial scales, local [80,81], subcontinental [82–84], continental and global [59].

## 6. The Term "Climate Change"

According to our proposal in Section 4, and given that the term process means change (as clearly stated by Kolmogorov [85,86], who introduced it in the scientific vocabulary), climate is changing by definition. Thus, there is no need to define or use the term climate change. Change occurs at all scales [33], and there is nothing particular in any specific one, like the commonly assumed 30-year scale. By studying long observation series of atmospheric and hydrological processes, one would see that the only characteristic scale with clear physical meaning is the annual; beyond that there is no objective "border scale" that would support a different definition of climate. The above definition includes all scales beyond the annual, thus leaving out the smaller scales (e.g., of several minutes or days) to be associated with weather.

Hence, in scientific terms, the content of the term climate change is almost equivalent to that of weather change or even time change (climate is changing as is weather and time).

Actually, the term climate change, which appeared in literature only after the 1970s (Figure 8), serves nonscientific purposes [87]. Quoting Vít Klemeš (from [88]) "the term 'climate change' is a misleading popular slogan".

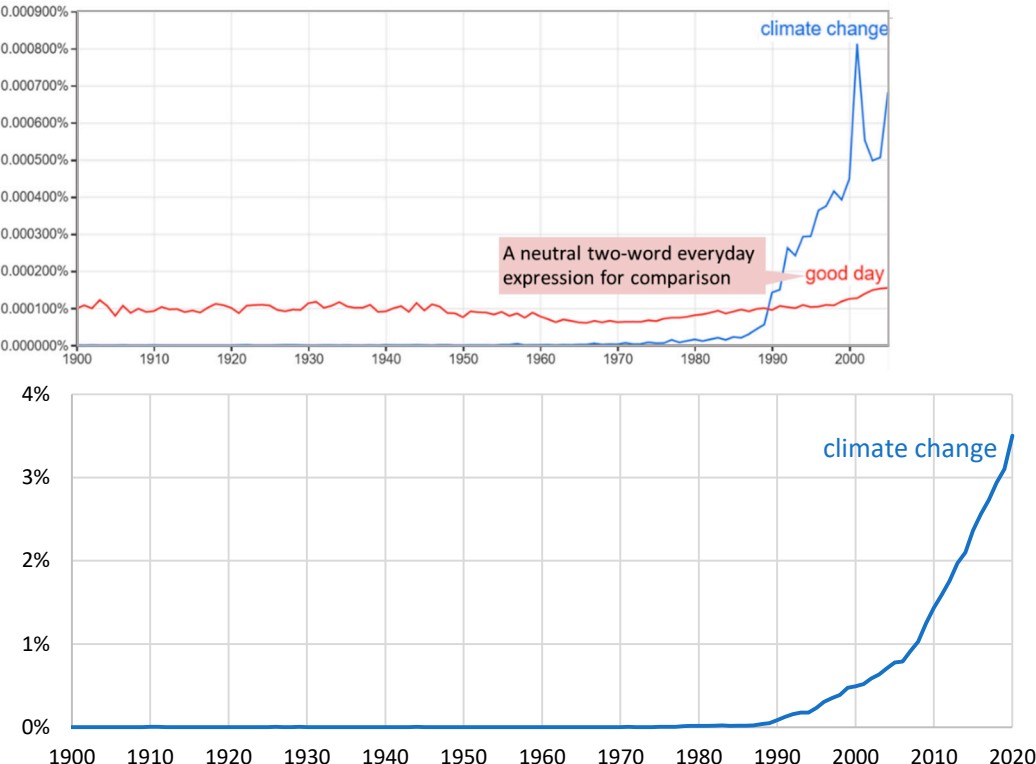

**Figure 8.** Evolution of the frequency of appearance of the term "climate change": (**upper**) in the millions of books archived by Google Books (the neutral term "good day" is used as a reference for comparison, and it appears that after 1990 "climate change" became more important than "good day"); (**lower**) in the 78 million items contained in the Scopus database of scientific articles (the search was conducted over "all fields").

Even according to the IPCC's [22] definition, the meaning of climate change is ambiguous:

*Climate change refers to a change in the state of the climate that can be identified (e.g., by using statistical tests) by changes in the mean and/or the variability of its properties, and that persists for an extended period, typically decades or longer. Climate change may be due to natural internal processes or external forcings such as modulations of the solar cycles, volcanic eruptions and persistent anthropogenic changes in the composition of the atmosphere or in land use.*

*Note that the Framework Convention on Climate Change (UNFCCC), in its Article 1, defines climate change as: 'a change of climate which is attributed directly or indirectly to human activity that alters the composition of the global atmosphere and which is in addition to natural climate variability observed over comparable time periods'. The UNFCCC thus makes a distinction between climate change attributable to human activities altering the atmospheric composition, and climate variability attributable to natural causes.*

The fact that climate change is a political, rather than a scientifically sound term is highlighted by several observations. One is the large number of the USA congressional hearings on climate change (Figure 9). Also, the earliest item in the Google Books collection, which includes the term "climate change" in its title, is a cold-war report by the CIA of 1976 [89] referring to the USSR (Figure 10). Interestingly, in it, climate change is meant as the cooling of the Northern Hemisphere since 1940. Specifically, the report states:

*the drought and subsequent famine in the Sahelian zone if North Africa during the late 1960s and early 1970s has focused world attention on the implication of climate change. According to evidence gathered by climatologists, the Northern Hemisphere has been cooling since the mid-1940s.*

It is also quite insightful in this respect to consult the Address to the Sixth Special Session of the United Nations General Assembly in April 1974 ([90] and Figure 10) by Henry Kissinger, the then Secretary of State and also National Security Advisor of the USA. Here he introduces climate change as an urgent problem and calls for immediate international action (a call that would become pretty regular thereafter):

*man-made disasters, have been threatened by a natural one: the possibility of climatic changes in the monsoon belt and perhaps throughout the world. The implications for global food and population policies are ominous. The United States proposes that the International Council of Scientific Unions and the World Meteorological Organization; urgently investigate this problem and offer guidelines for immediate international action.*

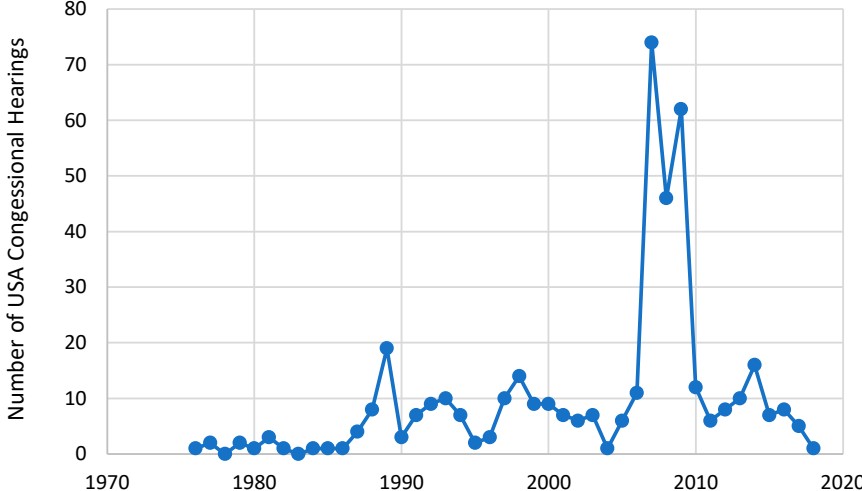

**Figure 9.** Annual number of USA Congressional Hearings on climate change, whose description contains either of the phrases "climate", "greenhouse" and "global warming"; the first instance appears in 1976 (data source: [91]; data coverage: 1946–2018).

And indeed, the World Meteorological Organization responded swiftly within a month with a report of its Secretary General entitled "Environmental Pollution and Other Environmental Questions–Implications of Possible Climatic Changes" [92,93]. This was followed by several actions and events by diverse American and international organizations, which, fourteen years after Kissinger's talk, resulted in the establishment of the IPCC in 1988 [94]. The efforts continue under that institution through the present day, while the political dimension of the efforts is highlighted by the accompanying activism, lately expanded to include schoolchildren.

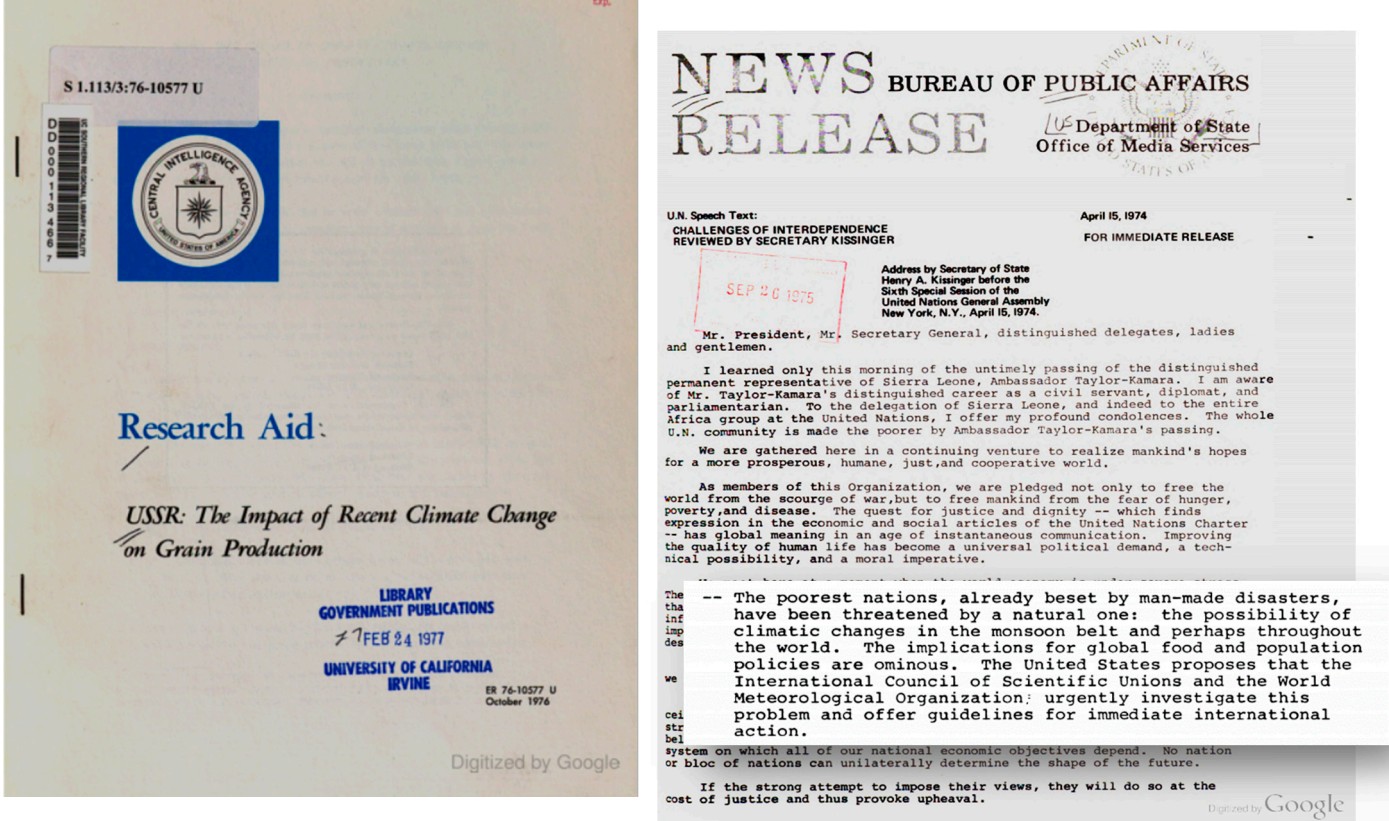

**Figure 10.** (**left**) The earliest item in the Google Books collection, which includes the term "Climate Change" in its title [89]. (**right**) Kissinger's (1974) Address to the Sixth Special Session of the United Nations General Assembly [90] with the inset showing the most interesting quotation about "climatic changes".

Interestingly, the same year, 1974, that Kissinger's speech calling for action was delivered, the Rockefeller Foundation (RF, with which Kissinger was a distinguished collaborator) announced perhaps the first in history conference whose title included the term "climate change". The announcement was included in the RF report of 1974 in this way [95]:

> ***Climate Change, Food Production, and Interstate Conflict***. *This interdisciplinary conference, organized jointly by RF officers from Conflict in International Relations, Quality of the Environment, and Conquest of Hunger programs, will bring together climatologists, scientists concerned with food production and others with experience with national public policy, and foundation representatives to examine the future implications of the global cooling trend now under way and its effects on world food production. Countries to be represented include the United States, Canada, the United Kingdom, and India.*

The same RF report is revealing in terms of the methods used by the RF to promote its agendas, including that of climate change:

> *Several science editors were asked to participate in Foundation meetings on climate change, food production and interstate conflict, genetic resistance in plants to pests, and aquaculture. Stories appeared subsequently on the front page of The New York Times, and the Associated Press carried substantial stories which were widely used. In each instance, the writers were introduced to our program officers and encouraged to use them as resource people. (Officers are now, in fact, being called on by journalists, particularly in areas of current high news interest such as food production, population problems, environmental issues, and the arts.)*

In the RF report of the next year (1975) [96], the same text of the conference announcement is repeated with a few differences, i.e., the future tense is changed to past tense, the phrase "the global cooling trend now under way" is changed to "the global cooling trend now currently underway" and the counties represented also included the Soviet Union, Japan and Germany. And indeed, the scheduled conference was held in June 1975 in Bellagio (located on Lake Como, Italy, where the Bellagio Center of RF is hosted) and, according to Hare [97], it:

> led to the publication in 1976 of a report called *Climatic Change, Food Production and Interstate Conflict*, which reached the desk of Henry Kissinger, then the U.S. Secretary of State.

A fascinating characteristic, evident in all documents of this early period before the establishment of IPCC, is that their authors are not confident whether climate change is natural or anthropogenic (see, e.g., the quotation by Kissinger above) or that their climate alert was about global cooling or global warming. Indeed, the answer was not categorical and in fact did not matter. What did matter was the alert per se. And it did not take much time to reverse the global cooling threat in the climate change agenda (see the quotation in the RF reports above) to global warming.

As climate change is a political issue, it is better to trace its development through a general-audience, rather than scientific, magazine. Thus, this reversal becomes evident by reviewing news articles in the Time Magazine:

- In 1974, an article entitled "Another Ice Age?" [98] stated:

> when meteorologists take an average of temperatures around the globe they find that the atmosphere has been growing gradually cooler for the past three decades. The trend shows no indication of reversing.

- In 1976, an article entitled "Environment: The World's Climate: Unpredictable" [99] gave two possibilities for the future climate. It is interesting to note that both are catastrophic and, while the two are opposite to each other, the intermediate case is not regarded a possibility at all:

> Climatologists still disagree on whether earth's long-range outlook is another ice age, which could bring mass starvation and fuel shortages, or a warming trend, which could melt the polar icecaps and flood coastal cities.

- In 1987, the front cover of the issue of 19 October announced "The Heat is On" and an article with the same title [100], mostly devoted to the depletion of ozone, also asserted:

> Potentially more damaging than ozone depletion, and far harder to control, is the greenhouse effect, caused in large part by carbon dioxide ($CO_2$). The effect of $CO_2$ in the atmosphere is comparable to the glass of a greenhouse: it lets the warming rays of the sun in but keeps excess heat from reradiating back into space. Indeed, man-made contributions to the greenhouse effect, mainly $CO_2$ that is generated by the burning of fossil fuels, may be hastening a global warming trend that could raise average temperatures between 2 degrees F and 8 degrees F by the year 2050.

- In 1989, less than a month after the establishment of the IPCC [94], the front cover of the issue of 2 January declared the "Endangered Earth" and the accompanying story [101] contained the following text, since then repeated myriad times (possibly rephrased but essentially with the same meaning):

> What would happen if nothing were done about the earth's imperiled state? According to computer projections, the accumulation of $CO_2$ in the atmosphere could drive up the planet's average temperature 3 degrees F to 9 degrees F by the middle of the next century. That could cause the oceans to rise by several feet, flooding coastal areas and ruining huge tracts of farmland through salinization. Changing weather patterns could make huge areas infertile or uninhabitable, touching off refugee movements unprecedented in history.

- In 1992, the salvation of the Earth began, as announced in the front-cover of Time's issue of 1 June: "Coming Together to Save the Earth" [102].

By now there is overwhelming evidence that "climate change" is a term in a political agenda (cf. World Economic Forum's *The Great Reset* [103], mixing climate change and Covid-19 to support the necessity for a great reset; Harari's *Homo Sapiens* [104] advocating the idea of a New Global Empire to solve the problem of accumulation of greenhouse gases; the joint report by the US National Intelligence Council & EU Institute for Security Studies for Global Governance [105,106]; the Rockefeller Brothers Fund's report *Change–Global Warming* [107]; and everyday news stories about diverse politico-ideological activist groups "fighting" climate change). Assuming that we have democracy, and freedom of opinion and speech, agreement or disagreement with any political agenda is any citizen's inalienable right. On the other hand, political agendas do not belong to the domain of science. History teaches that mixing up science with social aspects such as politics (cf. Eugenics and Lysenkoism) or religion (cf. Giordano Bruno and Galileo) has had tragic results both for science and society. Such mixing up has been admitted with pride by scientists who are proponents of the climate change agenda ([87]; Section 2) (also cf. "March for Science").

## 7. Discussion

Traditionally, hydrology has given emphasis on the information contained in data from observations and measurements, and followed an inductive scientific approach in order to make predictions. The predictions are of probabilistic character, based on the data and using stochastics. They have constituted the successful basis for engineering design and water management. Climatology has traditionally followed a similar path, by analysing long series of meteorological data. Both disciplines have been developed around the canon "in data we trust". Models have been regarded useful only if they were consistent with observations.

The reverse canon, giving more importance to models even if they disagree with observations, is epistemologically problematic. However, it becomes all the more widespread, particularly within climate research, as it better serves the set climate objectives and facilitates awarding of research grants. Nevertheless, this approach is not brand new, as exemplified by the following story [30,108]. In October 1941, Hitler's meteorologist Franz Baur issued a prediction that the next winter in Russia would be mild. However, that winter, which marked the Battle of Moscow as the first major Soviet counteroffensive of the war was committed, turned out to be one of the coldest in record. When reports by the military staff about the severe winter were communicated to Baur, his response was "the observations must be wrong".

The culture of working on virtual reality, and trusting models more than reality, has been very common in so-called climate science but has also affected hydrology. This has been the most severe impact of "climate change" on hydrology [109]. However, a warning about this problem was issued very early by Lamb [24]:

*The many ingenious formulations of theory of climate in terms of global mathematical models are likely to prove fallacious in one or another way unless and until equal attention is paid to establishing the past record of climatic behaviour and assimilating the lessons from it.*

A most important lesson from the past record is that climate has ever been changing. This has also been pointed out by Lamb [24], as quoted in Section 2, and stressed by other climatologists, such as Peixoto and Oort [110]:

*We know that climate has undergone many changes in the past and that it will continue to change in the future. In other words, the climate is always evolving and it must be regarded as a living entity. Thus we should avoid the misleading concept of the constant nature of climate.*

## 8. Conclusions

Given the hot and polarized discussions and actions about climate, it can be anticipated that many readers would find this paper useless, if not harmful. Actually, one of the aims of the paper is to show that polarization stems from political, rather than scientific, roots. Many scientists have paralleled their scientific profession with political aims (cf. "March for Science"). At the same time, mixing up science with politics has been promoted by many as a positive development. In contrast, this paper tries to promote the ancient ideal of science being separated from other interests, such as economic or political. It is recalled that Plato and Aristotle clarified the meaning and the ethical value of science as the pursuit of the truth; pursuit that is not driven by political and economic interests. For the latter, they used different terms, *sophist* (σοφιστής) and *sophistry* (σοφιστεία) [30,111–113].

In modern politics, fuzzy language and subjectivity may be desirable as they serve several purposes such as inclusiveness and diffusion of responsibility. In contrast, in science, the desiderata are rigour, clarity and objectivity. These desiderata may attribute some usefulness to this paper in clarifying concepts related to climate and water. Arguably, there is a strong need for such clarification if we accept that political influences should be left out.

Specifically, the current definitions of climate do not highlight its nonstatic nature. Rather, they imply a static climate, as already analysed (Section 3). Hopefully, the definition proposed and illustrated here (Section 4), which highlights the stochastic character of climate, could be useful to dispel this fallacy or, at least, provoke some discussion toward a more rigorous definition. By dispelling the fallacy, the term "climate change" would hopefully disappear from the scientific vocabulary and remain where it exactly belongs, i.e., the political vocabulary (Section 6). Dispelling another set of fallacies about the relationship of water and climate, also investigated here (Section 5) could be equally useful.

The potential usefulness relies on at least two facts. Highlighting the stochastic character of climate and its huge variability helps us understand the failure of current deterministic modelling approaches in describing past climate, and points to a potentially more promising direction in climate modelling within a stochastic framework. Highlighting the strong role of water in the climate can help shake the prevailing views on roles and causality chains in climatic processes, which may currently be opposite the real ones.

**Funding:** This research received no external funding but was motivated by the scientific curiosity of the author.

**Institutional Review Board Statement:** Not applicable.

**Informed Consent Statement:** Not applicable.

**Data Availability Statement:** All data used in the examples are available online and free to download from the sites given in the text.

**Acknowledgments:** I thank the guest editors Andreas Angelakis and Stavros Alexandris for the invitation to write this paper, as well as the guest editor Vasileios Tzanakakis for professionally handling the review process. I am grateful to two anonymous reviewers whose constructive comments helped me to improve the paper. I also thank a third anonymous reviewer for the careful reading of the paper, for suggesting detailed corrections, for the negative comments, which helped me to strengthen the paper against them, and for his (or her) generosity to give a positive recommendation in the second review round. I also acknowledge a fourth anonymous reviewer who wrote a summary of my paper in his (or her) own interpretation, followed by his (or her) opinions that "Climate change is necessary in science and politics" and that "the expected changes [in climate] are worrying". Finally, I thank Willis Eschenbach who helped me to locate the CERES data.

**Conflicts of Interest:** The author declares no conflict of interest.

## Appendix A. Ancient and Early Modern Quotations about Climate

Meteorological phenomena and, in particular, those related to the presence of water in the atmosphere, occupied all Greek natural philosophers who gave birth to science (6th cen-

tury BC) and laid the foundation of hydrological concepts and the hydrological cycle [114]. Aristotle streamlined the knowledge of the time in his book *Meteorologica*. The related terms *meteor* (μετέωρον, meaning raised from off the ground), *meteorology* (μετεωρολογία) and meteorologist (μετεωρολόγος) were used even before Aristotle (namely, by Plato and Euripides). However, the term *atmosphere* (ατμόσφαιρα, Latin: *atmosphaera*, meaning a sphere of vapour), while it is also of Greek origin, does not appear in the classical texts but most probably was coined in the 17th century AD [115].

The notion of climate is also old. Herodotus described the different climates of some areas but without penetrating into the nature of climate, for which he has not even proposed a particular term, but calls ouranos (οὐρανὸς, meaning *heaven*). He further connects the climate with human ethics and behaviours, sometimes adding elements, possibly by imagination, which are difficult to believe. In the following passages, he describes the climates of Egypt and Scythia, a region of Central Eurasia, west, east and north of the Caspian Sea (*The Histories*, 2.35 and 4.28, respectively):

*Αἰγύπτιοι ἅμα τῷ οὐρανῷ τῷ κατὰ σφέας ἐόντι ἑτεροίῳ καὶ τῷ ποταμῷ φύσιν ἀλλοίην παρεχομένῳ ἢ οἱ ἄλλοι ποταμοί, τὰ πολλὰ πάντα ἔμπαλιν τοῖσι ἄλλοισι ἀνθρώποισι ἐστήσαντο ἤθεά τε καὶ νόμους.*

*(The Egyptians in agreement with their climate, which is unlike any other, and with the river, which shows a nature different from all other rivers, established for themselves manners and customs in a way opposite to other men in almost all matters–English translation by G.C. Macaulay, 1890).*

*δυσχείμερος δὲ αὕτη ἡ καταλεχθεῖσα πᾶσα χώρη [Σκυθία] οὕτω δή τι ἐστί ἔνθα τοὺς μὲν ὀκτὼ τῶν μηνῶν ἀφόρητος οἷος γίνεται κρυμός, ἐν τοῖσι ὕδωρ ἐκχέας πηλὸν οὐ ποιήσεις, πῦρ δὲ ἀνακαίων ποιήσεις πηλόν: ἡ δὲ θάλασσα πήγνυται καὶ ὁ Βόσπορος πᾶς ὁ Κιμμέριος, καὶ ἐπὶ τοῦ κρυστάλλου οἱ ἐντὸς τάφρου Σκύθαι κατοικημένοι Στρατεύονται καὶ τὰς ἁμάξας ἐπελαύνουσι πέρην ἐς τοὺς σίνδους. οὕτω μὲν δὴ τοὺς ὀκτὼ μῆνας διατελέει χειμὼν ἐών, τοὺς δ' ἐπιλοίπους τέσσερας ψύχεα αὐτόθι ἐστί. κεχώρισται δὲ οὗτος ὁ χειμὼν τοὺς τρόπους πᾶσι τοῖσι ἐν ἄλλοισι χωρίοισι γινομένοισι χειμῶσι, ἐν τῷ τὴν μὲν ὡραίην οὐκ ὕει λόγου ἄξιον οὐδέν, τχονταιο δὲ θέρος ὕων οὐκ ἀνιεῖ: βρονταί τε ἦμος τῇ ἄλλῃ γίνονται, τηνικα ῦτα μὲν οὐ γίνονται, θέρεος δὲ ἀμφιλαφέες: ἢν δὲ χειμῶνος βροντὴ γένηται, ὡς τέρας νενόμισται θωμάζεσθαι. ὡς δὲ καὶ ἢν σεισμὸς γένηται ἥν τε θέρεος ἥν τε χειμῶνος ἐν τῇ Σκυθικῇ, τέρας νενόμισται. ἵπποι δὲ ἀνεχόμενοι φέρουσι τὸν χειμῶνα τοῦτον, ἡμίονοι δὲ οὐδὲ ὄνοι οὐκ ἀνέχονται ἀρχήν: τῇ δὲ ἄλλῃ ἵπποι μὲν ἐν κρυμῷ ἑστεῶτες ἀποσφακελίζουσι, ὄνοι δὲ καὶ ἡμίονοι ἀνέχονται.*

*(This whole land [Scythia] which has been described is so exceedingly severe in climate, that for eight months of the year there is frost so hard as to be intolerable; and during these if you pour out water you will not be able to make mud, but only if you kindle a fire can you make it; and the sea is frozen and the whole of the Kimmerian Bosporus, so that the Scythians who are settled within the trench make expeditions and drive their waggons over into the country of the Sindians. Thus it continues to be winter for eight months, and even for the remaining four it is cold in those parts. This winter is distinguished in its character from all the winters which come in other parts of the world; for in it there is no rain to speak of at the usual season for rain, whereas in summer it rains continually; and thunder does not come at the time when it comes in other countries, but is very frequent, in the summer; and if thunder comes in winter, it is marvelled at as a prodigy: just so, if an earthquake happens, whether in summer or in winter, it is accounted a prodigy in Scythia. Horses are able to endure this winter, but neither mules nor asses can endure it at all, whereas in other countries horses if they stand in frost lose their limbs by mortification, while asses and mules endure it–ibid.)*

Aristotle in his *Meteorologica* (362b.17) describes the relationship of climate with latitude:

*ὅ τε γὰρ λόγος δείκνυσιν ὅτι ἐπὶ πλάτος μὲν [τὴν οἰκουμένην] ὥρισται, τὸ δὲ κύκλῳ συνάπτειν ἐνδέχεται διὰ τὴν **κρᾶσιν**, -οὐ γὰρ ὑπερβάλλει τὰ καύματα*

*καὶ τὸ ψῦχος κατὰ μῆκος, ἀλλ' ἐπὶ πλάτος, ὥστ' εἰ μή που κωλύει θαλάττης
πλῆθος, ἅπαν εἶναι πορεύσιμον, —καὶ κατὰ τὰ φαινόμενα περί τε τοὺς πλοῦς καὶ
τὰς πορείας·*

*(theoretical calculation shows that* [inhabited Earth] *is limited in breadth, but could
as far as climate is concerned, extend round the Earth in a continuous belt; for it is not
difference of longitude but of latitude that brings great variation of temperature, and if
were not for the ocean which prevent it, the complete circuit could be made. And the facts
known to us from journeys by sea and land also confirm the conclusion;*

—English translation by H.D.P. Lee, Harvard University Press, Cambridge, MA,
USA, 1952).

Strabo in his *Geography* (1.1) uses the term climate, no longer identifying it with the latitude
but linking it to the temperature:

*πάντες, ὅσοι τόπων ἰδιότητας λέγειν ἐπιχειροῦσιν, οἰκείως προσάπτονται καὶ
τῶν οὐρανίων καὶ γεωμετρίας, σχήματα καὶ μεγέθη καὶ ἀποστήματα καὶ* **κλίματα**
*δηλοῦντες καὶ θάλπη καὶ ψύχη καὶ ἁπλῶς τὴν τοῦ περιέχοντος φύσιν.*

*(Everyone who undertakes to give an accurate description of a place, should be particular
to add its astronomical and geometrical relations, explaining carefully its extent, distance,
degrees of latitude, and 'climate'—the heat, cold, and temperature of the atmosphere.*

—English translation by H.C. Hamilton, and W. Falconer, M.A., 1903)

In *Geography* (2.3), Strabo defines the five climatic zones on Earth that are used even
today. Notice that the Aristotelian term *crasis* (κρᾶσις) survives through the term εὔκρατοι
(*temperate*) zones, a term that is still in use in modern Greek:

*αὕτη δὲ τῷ εἰς τὰς* [πέντε] *ζώνας μερισμῷ λαμβάνει τὴν οἰκείαν διάκρισιν: αἵ
τε γὰρ κατεψυγμέναι δύο τὴν ἔλλειψιν τοῦ θάλπους ὑπαγορεύουσιν εἰς μίαν
τοῦ περιέχοντος φύσιν συναγόμεναι, αἵ τε* **εὔκρατοι** *παραπλησίως εἰς μίαν τὴν
μεσότητα ἄγονται, εἰς δὲ τὴν λοιπὴν ἡ λοιπὴ μία καὶ διακεκαυμένη.*

*(In the division into* [five] *zones, each of these is correctly distinguished. The two frigid
zones indicate the want of heat, being alike in the temperature of their atmosphere; the
temperate zones possess a moderate heat, and the remaining, or torrid zone, is remarkable
for its excess of heat.*

—English translation by H.C. Hamilton, and W. Falconer, M.A., 1903).

The following definition that appears in Moxon (1700) does not differ substantially from
the ancient Greek definitions:

*Climate, From the Greek word Clima. of the same signification; it is a portion of the Earth
or Heaven contained between two Parallels. And for distinction of Places, and different
temperature of the Air, according to their situation; the whole Globe of Earth is divided
into 24 Northern, and 24 Southern Climates, according to the half-hourly encreasing
of the longest days; for under the Equator we call the first Climate: from thence as far
as the Latitude extends, under which the longest day is half an hour more than under
the Equator, viz. 12 h and an half, is the second Climate: where it is increased a whole
hour, the third Climate: and so each Northerly and Southerly Climate respectively hath
its longest day half an hour longer than the former Climate, till in the last Climate North
and South, the Sun Sets not for half a year together, but moves Circularly above the
Horizon.*

## Appendix B. Definitions of Climate in Modern Books

The definition of climate by the landmark book on climate by Lamb [13] has been
already quoted in Section 2. However, this definition appears to be inconsistent with
Lamb's [24] observation of an ever-changing climate, also quoted above (Section 3). By
scrutinizing books related to climate we may see a diversity of definitions, some complying
with the community definitions given in Section 3, but others being different.

We first quote the definitions of climate given in the most influential books about climate, as quantified by the fact that they received several thousands of citations (according to Google Scholar). Thus, according to the book *Tree Rings and Climate* by Fritts [116],

> Climate *may be defined simply as an expression of meteorological phenomena representing weather occurring over a long period of time. It includes processes of exchange of energy and mass between the earth and the atmosphere.* [ . . . ] *The aggregate of climatic conditions expressed as means, variance and extremes for a region over a period of many years is referred to as the* macroclimate*. The data used to describe the macroclimate are often based either upon 30 or more years of observation from a single station or upon the average of several measurements from stations at diverse locations within a particular region.*

Furthermore, the book asserts that macroclimate's spatial and temporal domain are regional and many years, respectively, and also uses the notions of *climatic state* and *microclimate*, with spatial domains regional and local, respectively, and temporal domain one month to a number of years and minutes to years, respectively. Interestingly the microclimate's definition makes it similar to weather, which is asserted to have a time domain of minutes to days.

In another celebrated book by Von Storch and Zwiers [117], entitled *Statistical Analysis in Climate Research*, climate is defined this way:

> *Description of the climate consisted primarily of estimates of its mean state and estimates of its variability about that state, such as its standard deviations and other simple measures of variability.* [ . . . ] *The main purpose of this description is to define 'normals' and 'normal deviations,' which are eventually displayed as maps.*

Yet another celebrated book, *The Climate Near the Ground* by Geiger, Aron and Todhunter [118] adopts the definition by Linacre [119] who, in turn, presents it as "a consensus of sixteen published definitions" without naming them. The latter definition reads:

> *Climate is the synthesis of atmospheric conditions characteristic of a particular place in the long term. It is expressed by means of averages of the various elements of weather, and also the probabilities of other conditions, including extreme events.*

Interestingly, by looking at Table 1-1 of the book by Geiger et al., one sees that they assign values of what they call "primary time scale" to different instances of climate. These disagree with the common definitions of climate including their own. Namely, they specify the primary time-scale for microclimate to <10 s and even for the macroclimate they give a time scale between $10^5$ and $10^6$ s, i.e., 1 to 12 d.

A similar situation is met in the book *Boundary Layer Climates*, by Oke [120], who, without providing a particular definition of climate, makes it clear (in his Figure 1.1) that he refers to time scales between 1 and $10^5$ s (1 s to 1 d), which would normally be regarded as the time scale of weather rather than of climate.

A final quotation from the category of highly cited books on climate, namely from the book *Physics of Climate* by Peixoto and Oort [110], is quite useful:

> *We will regard the climate in a very broad sense in terms of the mean physical state of the climatic system. The climate can then be defined as a set of averaged quantities completed with higher moment statistics (such as variances, covariances, correlations, etc.) that characterize the structure and behavior of the atmosphere, hydrosphere, and cryosphere over a period of time. This definition of climate includes the more narrow traditional concept of climate based on the mean atmospheric conditions at the earth's surface.*

For completeness, it is useful to also review books on atmospheric science. Thus, in their highly cited book *Atmospheric Science*, Wallace and Hobbs [121] define climate thus:

> *the long term statistical properties of the atmosphere* [ . . . ] *constitute climate (for example, mean values and range of variability of various measurable quantities, such as temperature, and the frequencies of various events, such as rain or high winds, as a function of geographical location, season and time of day).*

Likewise, Andrews [122] in his book *An Introduction to Atmospheric Physics* states:

*The word climate refers to the state of the atmosphere on longer time scales, typically averaged over several years or more. The understanding of climate and climate change does not necessarily require a complete understanding of every weather event; conversely there are physical processes, operating on long time scales, that are unimportant for weather prediction but crucial for climate prediction. (An example is heat transport from deep ocean which may vary on decadal or longer time scales.)*

Finally, in his book *Essentials of Meteorology: An Invitation to the Atmosphere*, Ahrens [123] defines climate as follows:

*If we measure and observe these weather elements over a specified interval of time, say, for many years, we would obtain the "average weather" or the climate of a particular region. Climate, therefore, represents the accumulation of daily and seasonal weather events (the average range of weather) over a long period of time. The concept of climate is much more than this, for it also includes the extremes of weather—the heat waves of summer and the cold spells of winter—that occur in a particular region. The frequency of these extremes is what helps us distinguish among climates that have similar averages.*

## Appendix C. Examples of Climatic Evolution for Maximum and Minimum Daily Temperature

The GHCN-D database contains 15 stations with observations of daily maximum and minimum temperature and length greater than 150 years, most of which are located in Europe. We prefer to analyse maximum and minimum temperature because these are more objectively and accurately measured, while the daily average temperature is calculated by a number of observations within a day and with a particular algorithm, both of which may be subject to change in the course of time. For our examples, among the 15 stations (whose data sets are readily available for download from the ClimExp platform [31]), we choose.

- for maximum daily temperature: Vienna (Wien, Austria; 48.23° N, 16.35° E, 199.0 m; GHCN-D station code: AU000005901; WMO station: 11035), covering the period 1855–2020 (167 years of data), and
- for minimum daily temperature: Melbourne (Melbourne Regional Office, Australia; 37.81° N, 144.97° E, 31.0 m; GHCN-D station code: ASN00086071; WMO station: 94868), covering the period 1855–2014 (160 years of data).

The data are plotted on annual and climatic scales in Figure A1 (Vienna) and Figure A2 (Melbourne), in a manner similar to Figures 4 and 6 (Bologna precipitation). It can be seen that the maximum temperature (Vienna) has a behaviour similar to precipitation, exhibiting long-term fluctuations. The minimum temperature shows a different pattern, with an almost monotonic trend, throughout the entire period. This monotonic trend of minimum temperature (which is rather favourable for human life) has been observed also in other cities [30] and could partly be related to urbanization, but is also a more general behaviour worldwide [124].

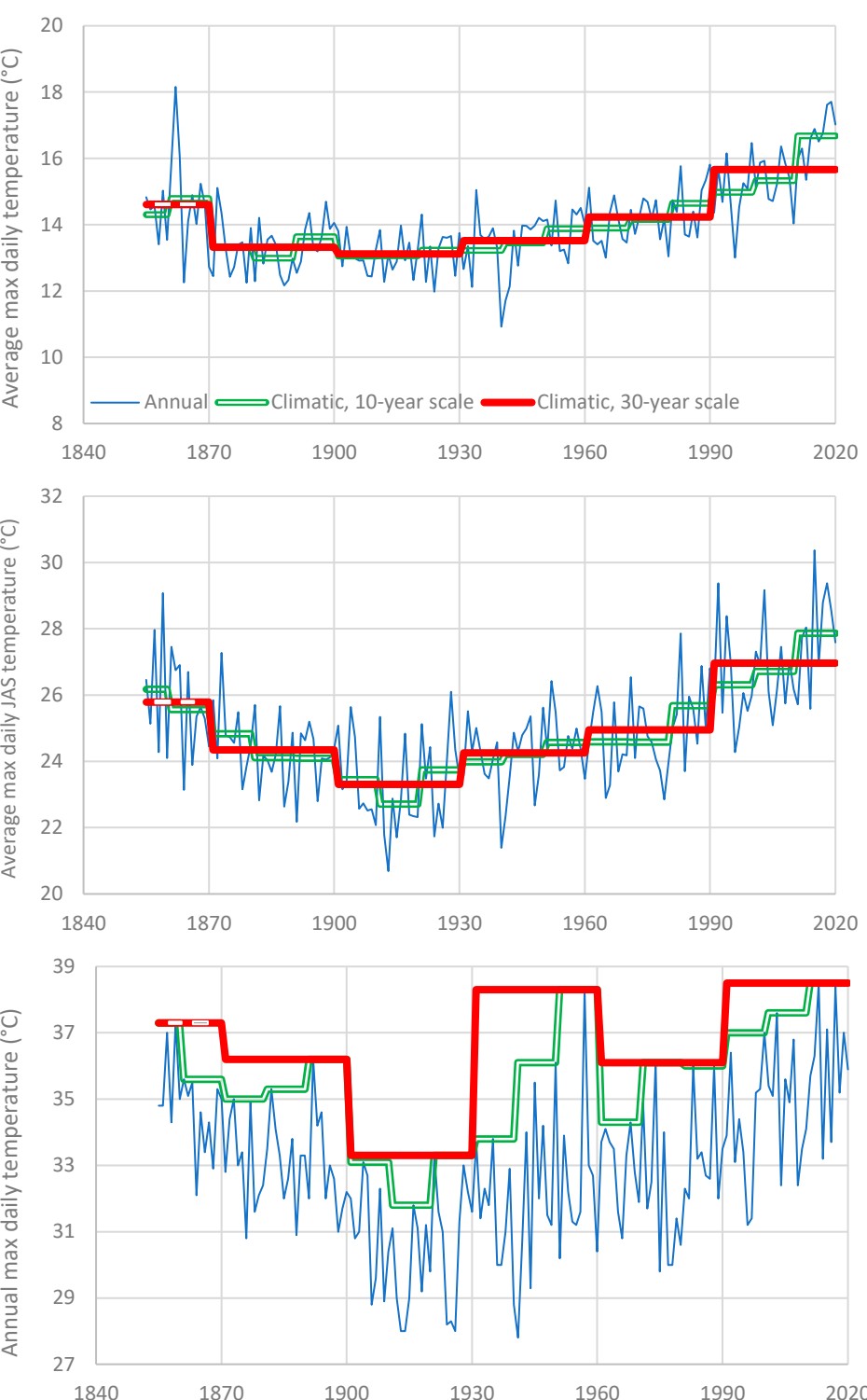

**Figure A1.** (**upper**) Evolution of annual average of maximum daily temperature in Vienna, as a climatic element, seen at the annual and the climatic time scales of 10 and 30 years; (**middle**) as in upper panel but for a time window of the three summer months, JAS; (**lower**) as in upper panel but for the annual maximum of maximum daily temperature.

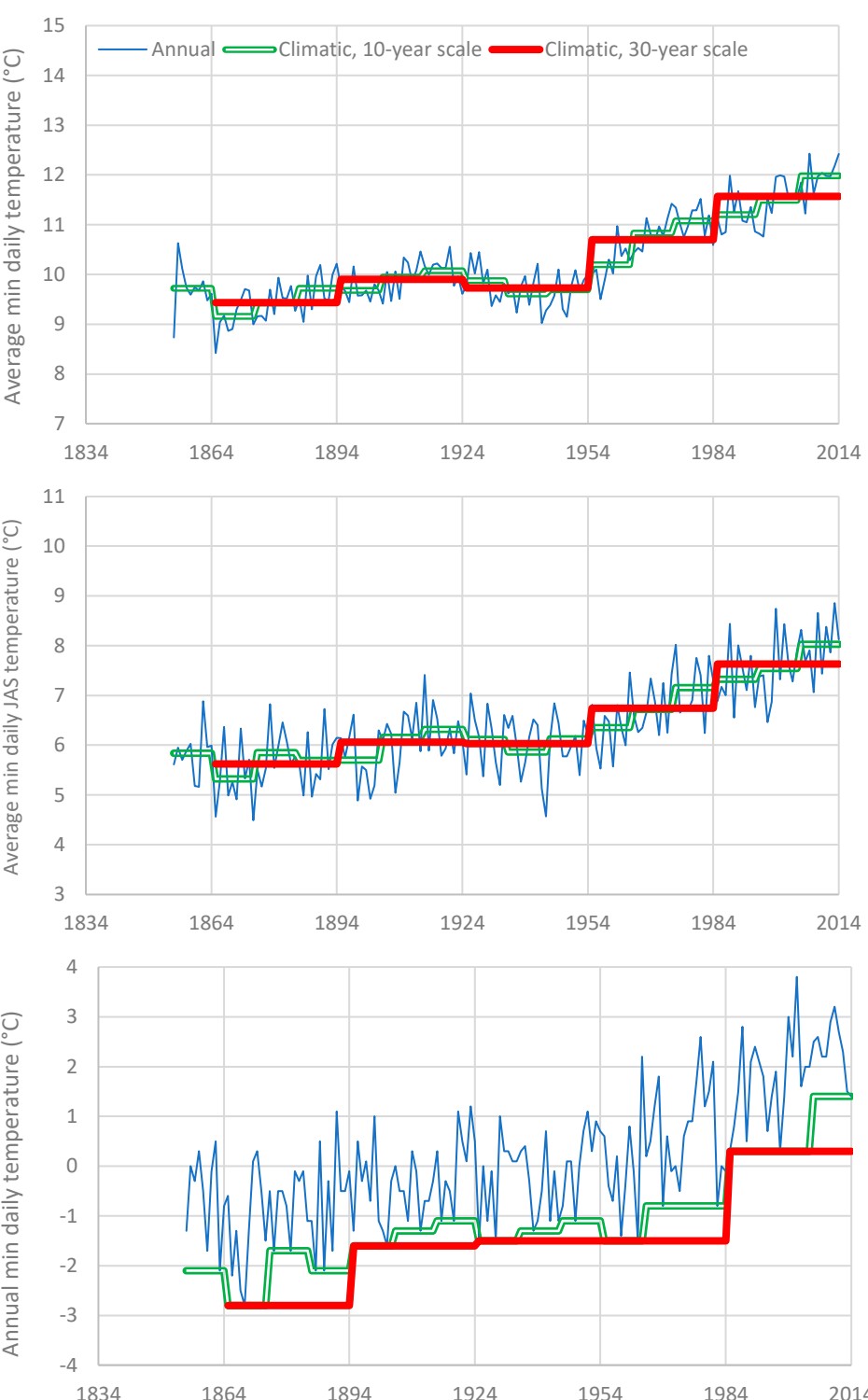

**Figure A2.** (**upper**) Evolution of annual average of minimum daily temperature in Melbourne, as a climatic element, seen at the annual and the climatic time scales of 10 and 30 years; (**middle**) as in upper panel but for a time window of the three summer months, JAS; (**lower**) as in upper panel but for the annual minimum of minimum daily temperature.

The climacograms in the two cases are shown in Figure A3. In comparison to the climacogram of precipitation in Bologna (Figure 5), the main difference here is the prominent seasonality. However, this is easily modelled by adding a harmonic component

(Equation (10)) to the FHK model (Equation (7)). As shown in Figure A3, this model describes very satisfactorily the empirical climacograms.

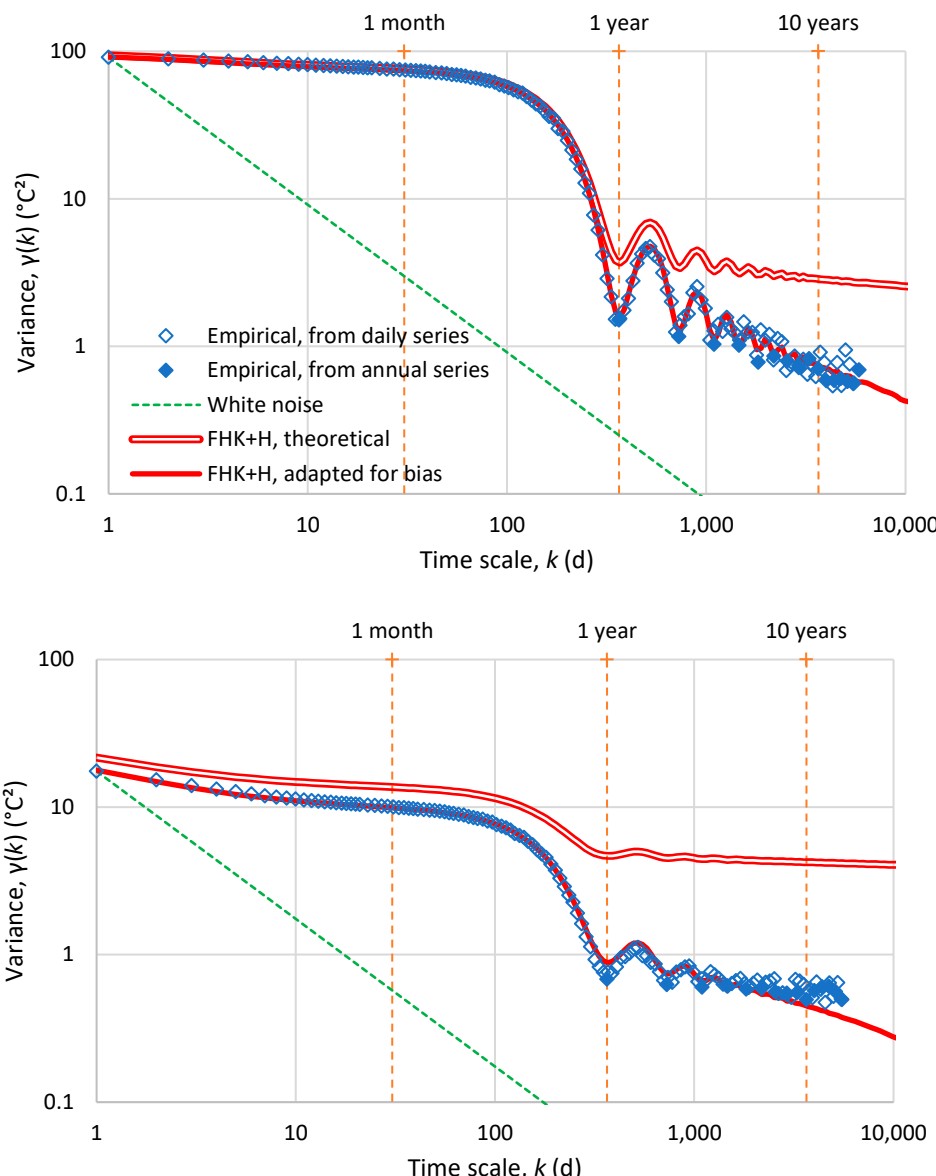

**Figure A3.** Climacograms of (**upper**) maximum daily temperature in Vienna and (**lower**) minimum daily temperature at Melbourne. FHK+H stands for filtered Hurst-Kolmogorov (FHK) process with an added harmonic oscillation (H). The climacogram of the FHK process and the harmonic have the mathematical expressions of Equations (7) and (10), respectively. The fitted parameters are for Vienna $H = 0.95$, $M = 0.95$, $\alpha = 1.50$ d, $\lambda_1 = 6.12$ °C$^2$, $\lambda_2 = 20.31$ °C$^2$, $\lambda_T = 70.97$ °C$^2$ and for Melbourne $H = 0.98$, $M = 0.98$, $\alpha = 0.38$ d, $\lambda_1 = 6.08$ °C$^2$, $\lambda_2 = 15.91$ °C$^2$, $\lambda_T = 8.36$ °C$^2$.

## Appendix D. Rough Calculation of Earth's Energy Imbalance

The recent energy imbalance in the climate system, or the net absorbed energy by the Earth, according to Trenberth et al. [68], is estimated at 0.9 W/m$^2$. Similarly, in a very recent study Cheng et al. [125] state:

*Currently there is an energy imbalance in the Earth's climate system of almost 1 W m$^{-2}$ [. . . ] Over 90% of this excess heat is absorbed by the oceans, leading to an increase of ocean heat content (OHC) and sea level rise, mainly through thermal expansion and melting of ice over land.*

The ocean's contribution in absorbing excess heat is estimated by IPCC (Box 3.1, Figure 1 in [22]) for the period 1970–2010 somewhat higher than 90%, namely 94% (63% for the upper ocean and 31% for the deep ocean) with the remaining 6% distributed among ice (2.5%), land (2.5%) and atmosphere (1%).

Interestingly however, Cheng's et al. analysis in the same study [125] finds:

*The new results indicate a total full-depth ocean warming of 380 ± 81 ZJ (equal to a net heating of $0.39 \pm 0.08$ W $m^{-2}$ over the global surface) from 1960 to 2020, with contributions of 40.3%, 21.6%, 29.2% and 8.9% from the 0–300-m, 300–700-m, 700–2000-m, and below-2000-m layers, respectively.*

This means that the energy imbalance in the Earth should be estimated at 0.39/0.9 = 0.43 W/m$^2$ (or 0.39/0.94 = 0.41 W/m$^2$), rather than 1 W/m$^2$.

To resolve this contradiction, we cross-check these figures by our own calculations based on [48]. According to [48], the increase of heat stored in the oceans in the last 50 years, measured at a 10-year climatic scale, is 277 ZJ or 5.5 ZJ/year. Considering the contribution of the other parts, as discussed above, we increase this value rounding it to 6 ZJ/year (an applied increase of 8%). By converting ZJ to J and year to s, and dividing with the area of the Earth ($5.101 \times 10^{14}$ m$^2$) we find a value of Earth's radiative imbalance of 0.37 W/m$^2$, close to the value 0.43 (or 0.41) W/m$^2$ resulting from Cheng et al. and far from the more commonly assumed value 0.9 (or 1.0) W/m$^2$.

**Appendix E. Earth's Albedo Changes**

In contrast to the inundation of literature, formal and informal, by plots of time-series of the global temperature, a literature review showed that there is little information on the climatic information of the Earth's albedo in absolute terms. A few papers and web sites give some "anomalies" on the monthly scale; for example, the site Measuring Earth's Albedo by NASA [126] shows a graph with annual "anomalies" for the period 2000–2011; this is also the case with a recent publication by Zhan et al. [127]. One of the rare exceptions is a publication by Loeb and Wielicki [128], which gives a time-series plot of the top of atmosphere (TOA) global albedo but, unfortunately, only for one year, 2010. For this reason (and after a reviewer's suggestion to include multidecadal data on albedo) we process shortwave radiation data from two sources:

The first data set is provided by NASA through its Giovanni platform [129] for the period 1980–2020. Specifically, we use data from the Modern-Era Retrospective analysis for Research and Applications version 2 (MERRA-2), a NASA atmospheric reanalysis for the satellite era using the Goddard Earth Observing System Model, Version 5 (GEOS-5) with its Atmospheric Data Assimilation System (ADAS), version 5.12.4 [130]. The specific variables used from this dataset are:

1. TOA incoming shortwave flux (M2TMNXRAD.5.12.4, W/m$^2$);
2. TOA net downward shortwave flux, which is the difference between incoming and outgoing shortwave radiation from the Earth's surface (M2TMNXINT.5.12.4, W/m$^2$);
3. Cloud albedo as provided by NASA's MERRA-2 International Satellite Cloud Climatology Project (ISCCP) (from the COSP Satellite Simulator, M2TMNXCSP.5.12.4);
4. ISCCP total cloud area fraction, estimated as the number of cloudy pixels divided by the total number of pixels (M2TMNXCSP.5.12.4).

We produce a time series of the global TOA albedo by integrating the data series 1 and 2 over the globe for each month, taking the ratio of the data series 2 over 1 and subtracting from 1. This time series is plotted in the upper panel of Figure A4 at the original monthly scales as well as at the annual and 10-year climatic scale.

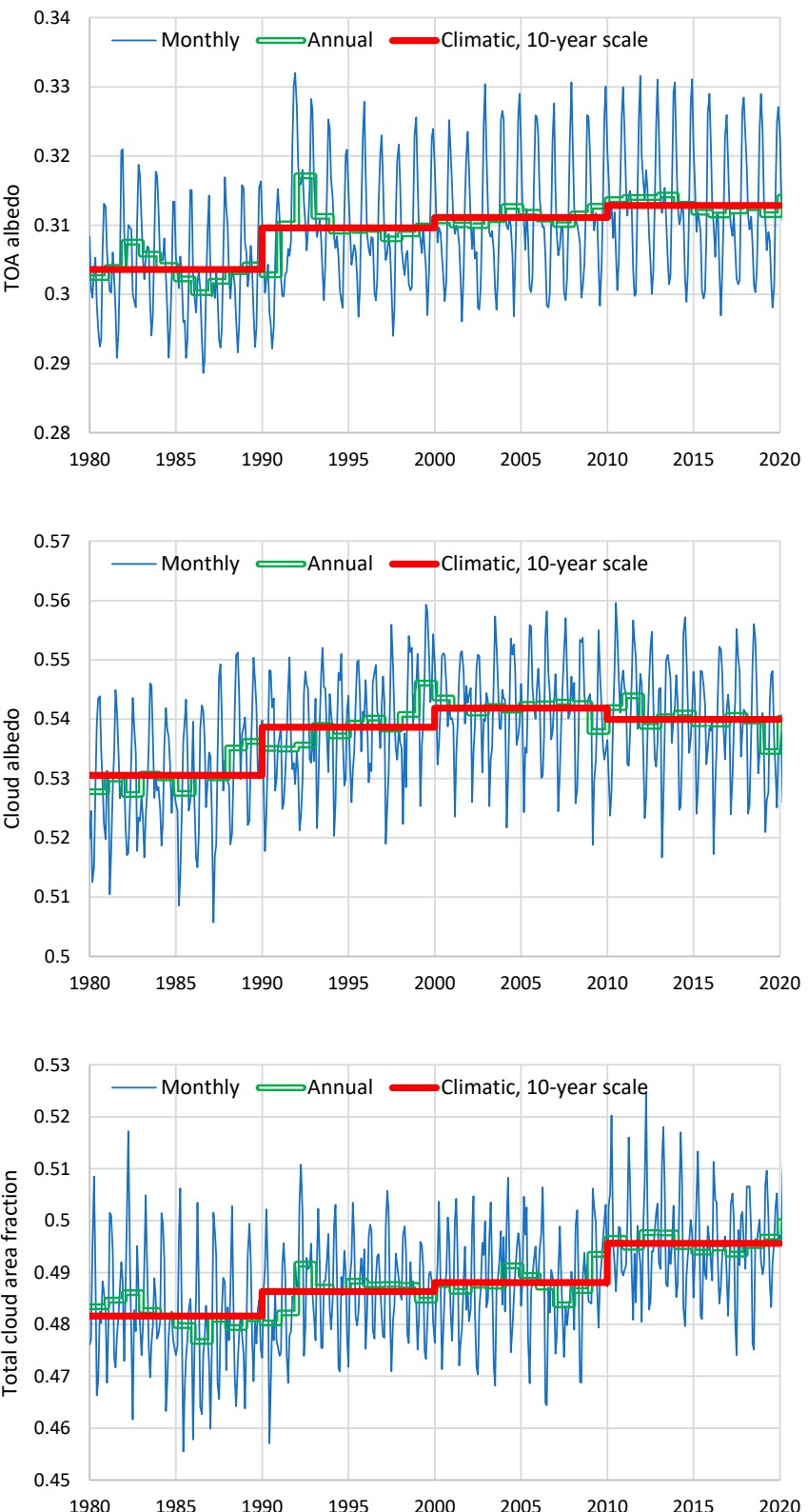

**Figure A4.** Time series of (**upper**) TOA albedo, (**middle**) cloud albedo and (**lower**) total cloud area fraction, as provided by NASA's MERRA-2 International Satellite Cloud Climatology Project (ISCCP).

The cloud albedo is readily given from data series 3, integrated over the globe, and is plotted in the middle panel of Figure A4. For completeness we have also plotted in the lower panel of Figure A4 the time-series of the total cloud area fraction integrated over

the globe. All three series plotted in Figure A4 suggest prominent variability, while their climacograms (not shown) suggest a high Hurst parameter of the order of 0.9, but the available time-series lengths are too short for a reliable estimation of this parameter. A useful quantitative estimate from these 40 years of data is that the range (maximum minus minimum) of each of the time series is about 0.02 for the annual scale and 0.01 for the 10-year climatic scale.

A final observation from Figure A4 is that the cloud albedo suggests fluctuating behaviour, consistent with the HK dynamics, while the other two series indicate an increasing trend throughout the 40-year period 1980–2020. Again, such a monotonic trend could be attributed to the short length of the time-series. While the observed increasing trends are consistent with each other (increasing cloud cover results in increasing TOA albedo), they appear inconsistent with the observed increase of global temperature.

The second time-series is from the Clouds and the Earth's Radiant Energy System (CERES), which is an on-going project using scientific satellite instruments. Again, this is part of the NASA's Earth Observing System, designed to measure both solar-reflected and Earth-emitted radiation from the top of the atmosphere (TOA) to the Earth's surface. The data used are from the TERRA platform for the monthly time scale and are available online [131] for the period March 2000 to November 2020. The global TOA albedo was calculated as the ratio, for each month, of the globally integrated observed TOA shortwave flux (all-sky) over the globally observed TOA solar insolation flux.

The CERES time series is plotted in the upper panel of Figure A5 at the original monthly scales, as well as at the annual and 10-year climatic scale. Again, there are fluctuations, which notably do not harmonize with those of MERRA-2. This is better shown in the lower panel of Figure A5, where the two time series are compared at the annual scale, along with their linear trends fitted on the period 2000–2020. Having made a shift in the plot of the two time series by 0.015 (see left and right vertical axes), in the first few years the two series agree but in later years they diverge. Furthermore, the linear trends are opposite to each other, one positive and the other negative. Also, the averages over the whole period disagree, being 0.283 and 0.312 for the CERES and the MERRA-2 series, respectively (note that Trenberth et al. [68] had suggested a value 0.30, which is in between the two).

All these are indications that the uncertainty in closing Earth's energy balance is still high. In addition, these time series do not fully describe the entire radiative balance of Earth, as they only refer to the shortwave radiation. The effect of the cloud cover on the longwave radiation is expected to be higher than in the shortwave radiation. However, the scope of this Appendix is not to study the radiative balance of Earth, but just to give an indication of the climatic variability of one of its components and the relationship of this variability with atmospheric water in the form of clouds.

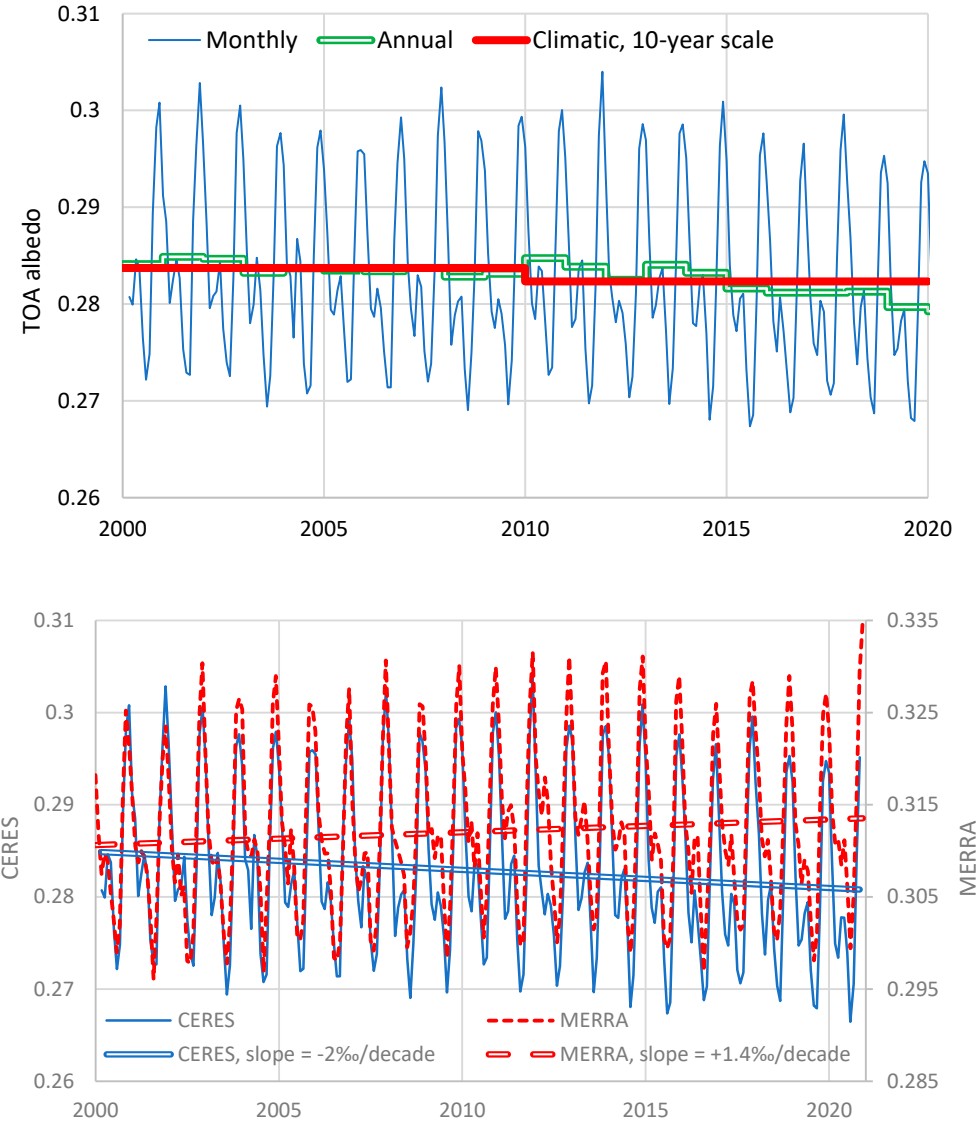

**Figure A5.** (**Upper**) TOA albedo time series, as provided by NASA's Clouds and the Earth's Radiant Energy System (CERES); (**lower**) the same series on annual time scale in comparison of the MERRA-2 series (Figure A4, upper panel) along with linear trends plotted for both series with fitted slopes shown in the legend.

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
