# Peer review of "Rethinking Climate, Climate Change, and Their Relationship with Water"

_water, doi:10.3390/w13060849_

Round 1

Reviewer 1 Report

This is an interesting submission addressing the important role of water in the maintenance and variability of climate. As an overall comment a significant proportion of the text seems to wander and should be tightened up. A number of comments are vague, and unsubstantiated. Many of the remarks refer to well-known phenomena and are not really relevant to the arguments presented here. Paper would benefit from a clearer narrative pathway and concrete suggestions.

Among some other things that struck me was that there was no mention of the ‘Anthropocene’, and the vast rigorous literature on this now (e.g., Daniel Buschmann, 2021: What is critical in the Anthropocene? A discussion of four conceptual problems from the environmental-political philosophy perspective. Ethics and Bioethics (in Central Europe), 10, 190-202, doi: 10.2478/ebce-2020-0018).

In a similar vein no mention was made that the present levels of atmospheric CO2 are highest over the last 1 Ma, and almost certainly since the Miocene, and have occurred in less than 200 years.

In addition to these I point to specific issues which must be addressed before I could recommend this manuscript for publication:

Lines 445-452:

Make clear that 341 W/m**2 is So / 4, and where this number come from. Also what is relevance of this statement on the seasonal cycle of albedo? If one were to argue that the multidecadal Earth’s Energy Imbalance (EEI) is due to an overall reduction in albedo (surface or planetary?) some independent (solar-weighted) albedo data should be used to support the implied argument here.

Lines 454-470: I found some of the information presented in this subsection as very misleading and confusing. Firstly, the article by Gavin Schmidt (Reference 47 - Gavin A. Schmidt, Reto A. Ruedy, Ron L. Miller and Andy A. Lacis, 2010: Attribution of the present-day total greenhouse effect. Journal of Geophysical Research, 115, D20106, doi: 10.1029/2010JD014287) was published eleven years ago and could hardly be regarded as ‘current’. More importantly, it has been appreciated for decades and longer that the strongest Greenhouse gas is in fact water vapor (e.g., see any basic text on atmospheric radiation or refer to any of the IPCC documents). As stressed in these studies, and indeed explicitly in Gavin’s paper, ‘In a doubled CO2 scenario … the magnitude of the total greenhouse effect is significantly larger than the initial radiative forcing, underscoring the importance of feedbacks from water vapor and clouds to climate sensitivity.’ That is, it is these feedbacks which introduce this multiplicity. A much more rigorous statement of the evidence and the physics is required here. Very valuable to refer in the paper to some recent studies pointing to the role of water vapor (and clouds) in warming the Arctic environment – some that should be referenced here are Lee, Feldstein et al., 2017: Revisiting the cause of the 1989-2009 Arctic surface warming using the surface energy budget: Downward infrared radiation dominates the surface fluxes. Geophys. Res. Lett., 44, 10,654–10,661,

Luo, and co-authors, 2017: Atmospheric circulation patterns which promote winter Arctic sea ice decline. Env. Res. Lett., 12, 054017, doi: 10.1088/1748-9326/aa69d0,

Screen, J. A., et al., 2018: Polar climate change as manifest in atmospheric circulation. Current Climate Change Reports, 4, 383-395, doi: 10.1007/s40641-018-0111-4.

Also, Demetris comments in this paragraph that ‘… the atmospheric CO2 … product of human emissions, contribute … only 3.8% to the global carbon cycle’ and quotes his Moscow lecture notes [Reference 48] on this. In reality, during the Holocene the carbon budget has been very close to balanced, made up of massive alternating seasonal fluxes from the atmosphere to the ocean and biosphere, associated with the seasonality of the biology and chemistry. The relatively small amount of anthropogenic activity is the factor which disturbs this balance. Valuable to refer to the ‘Keeling Curve’ (https://keelingcurve.ucsd.edu/), and the vast literature related to it.

In connection with the comment related to reference 50, I could not find this paper in the journal ‘Science’.

Lines 851- (Appendix C): The material presented here refers to the authors lecture notes. A much better (and refereed!) article to cite here would be …

Lijing Cheng, John Abraham, Kevin E. Trenberth, John Fasullo, Tim Boyer, Ricardo Locarnini, Bin Zhang, Fujiang Yu, Liying Wan, Xingrong Chen, Xiangzhou Song, Yulong Liu, Michael E. Mann, Franco Reseghetti, Simona Simoncelli, Viktor Gouretski, Gengxin Chen, Alexey Mishonov, Jim Reagan and Jiang Zhu, 2021: Upper ocean temperatures hit record high in 2020. Advances in Atmospheric Sciences, doi: 10.1007/s00376-021-0447-x.

Also, see my earlier comment on presenting solid evidence on a reduction of albedo by this amount, rather than just suggesting it is a possibility.

Reviewer 2 Report

The article by D. Koutsoyiannis "Rethinking climate, climate change, and their relationship with water" is, quite simply, an opinion or editorial piece and most certainly not original experimental research. At that, it remains grossly incomplete given the current state of knowledge in hydrology, water resources, and their relationships with climate.

First, the author uses this article for an absurd quantity of self-citations.

The translations from Greek that the author provides are vaguely interesting, but including the original Greek text is unnecessary.

Figure 1 is superfluous.

Figure 8 is misleading: Google n-gram searches include only books (not scientific journal literature), and only those books scanned and indexed by Google up to the time of the search. A better search would include Web of Science and other scientific journal databases.

The author proposes to show "that water is the main element that drives climate, rather than just being affected by climate as commonly thought" [lines 43-44]. A number of physical and thermodynamic characteristics of water in all phases are quoted in Section 5. However, these do not complete such a demonstration. Climate is, in my learning and view, driven by numerous elements: solar input, axial tilt, planetary rotation, land/ocean configurations and characteristics, and atmospheric composition, including water vapor. The author addresses almost none of these in order to demonstrate their minor contribution compared with water. In fact, the author seems to render many of these as "externalities" rather than integral components of the climate system. Declaring water as the principal driver of the climate system leaves out many larger influences in a questionable approach that the author's argument fails to justify.

Substantively, and of great consequence to the author's thesis, there is no mention in the manuscript of agriculture (other than the citation of a CIA report on Russian grain production), stationarity (other than in the title of a single reference), and the relation of those to water resources management. The current fundamental touchstone reference on stationarity, by Milly et al. [2008, Science], is not included here. The societal (and somewhat scientific) concept of climate stationarity is fundamentally intertwined with the establishment of perennial agriculture and the advent of urbanization more than 6Kya. With more sedentary lifestyles and less far-ranging seasonal migration, people developed water resource management strategies that, in a feedback loop, further reinforced the entrenchment of agricultural and urban systems. This feedback loop was supported by relatively stable (stationary) climate patterns over the past several millennia, to the point that sudden changes in those patterns today are creating unexpected stresses on, and threatening the continued viability of, those systems. None of these were established as political systems, though the author prefers to stress that "climate change" (which refers to those recent sudden changes in consistency and stationarity and does not deny that climate indeed has changed before, just not as suddenly in the span of human expectation) is an inherently and purely political construct. The author's argument is fundamentally flawed, primarily by neglecting the wider scientific, social, and political contexts in which "climate change" is embedded at this stage in human history.

Finally, some line-by-line notes:

line 43: "show" (not "sho")

line 100: "It comprises not only those conditions that can obviously 'near average' or 'normal' ..." (check grammar against original source)

line 292: "it" (not "in")

line 384: the "hot half-year" in Bologna, Italy, is in the AMJJAS period (the periods are reversed)

line 416: "larger" (not "smaller")

line 417: "respectively)" (not "respectively_")

line 438: "known as" (not "or else")

line 472: "ENSO, AMO, and IPO" should each be defined (but references are not likely necessary)

line 507: "IPCC" (not "IPPC")

line 528: "in its" (not "it is")

line 553: the process was not "concluded with the establishment of the IPCC in 1988" (the process continues even today). However, one might say that the process following Kissinger's speech "culminated in the establishment of the IPCC" with efforts continuing under that institution through the present day.

line 567: "in its" (not "it is")

line 579: "In the 1975 RF report..." (not "In next year's RF report...")

line 606: "ice caps" (not "icecaps") (confirm with original source)

lines 627-8: this is not a complete sentence; also, what is "the Earth salvation"?

lines 641-2: "commended" or "condemned"? One would think that scientists would condemn (not commend) such "mixing up," although here the author's point might be clarified by explaining what is meant by "the climate change agenda."

lines 730-1: I am certainly not a scholar of the Greek language, but I think the word translated as "prodigy" might be better translated in context (on the English side) as "portent" (they are synonyms but with different connotations)

line 799: "days" (not "dates")

line 821: This "final quotation" is then followed by three additional quotations

line 836: "... the state of the atmosphere longer timescales..." (check grammar against original source)

line 838: "... conversely there physical processes..." (check grammar against original source)

line 853: "or" (not "of")

Reviewer 3 Report

In this study, the author includes a historical review of the notion of climate, modern definitions of climate, a new definition of climate that includes the hydrosphere and stochastics processes, an analysis of the relationship between climate and water, and finally the author argue that the term “Climate Change” is political and shouldn’t be used by the scientific community.

I see the paper as a review of concepts instead of a novel approach. In my opinion the main topics of the paper are:

  • A new definition of climate that includes the hydrosphere and stochastics processes for a range of time scales.

However the relationship between hydrosphere and climatology is well known. In the same way, the climate as a stochastic process has been studied before.

  • The suggestion that the term “climate change” disappear from the scientific vocabulary.

The author argues that climate is always changing…But it is changing in the same manner? Should we use the term “Change in the climate change”? I think the term “Climate change” is necessary in science and politics. In the last decades the observed changes in climate are unprecedented and the expected changes are worrying.

Reviewer 4 Report

I must admit that I cannot remember when and where I have read such interesting and quality paper. Very original, I must say! Despite this, I am proposing a major revision. Only one thing is missing. This is a comparison with other locations in case study, i.e. Bologna case. If authors will do this, I will not have nothing to say, and this will be one of my ''soft & easy'' reviews.

Round 2

Reviewer 1 Report

I thank the author for his responses to my comments. There needs to be further clarification in the paper.

In my review of the original submission I commented that ‘As an overall comment a significant proportion of the text seems to wander and should be tightened up. A number of comments are vague, and unsubstantiated’. The author has responded with ‘However, being an old man now, I have developed through the years a style of writing which expresses myself. Some like it (cf. Reviewer 4), some not, but my personal taste is to avoid a stereotypical stylized text. I hope the Reviewer can tolerate that.’ This did not really address my point as to the structure of a scientific and rigorous paper. The guidelines for the ‘Water’ journal specify that ‘Water has no restrictions on the length of manuscripts, provided that the text is concise and comprehensive’. I will leave it to the Editor to decide whether the text is ‘concise’ and appropriate for the journal.

Another response I had some difficulty with was in connection with my comments on atmospheric CO2 levels since the Miocene and also the last ~1 Ma. Important that the author updates himself with the relevant literature on this issue. For a start, I recommend reading (and making reference to)

Ying Cui, Brian A. Schubert and A. Hope Jahren, 2020: A 23 m.y. record of low atmospheric CO2. Geology, 48, 888-892, doi: 10.1130/G47681.1 and

Gavin L. Foster, Dana L. Royer and Daniel J. Lunt, 2017: Future climate forcing potentially without precedent in the last 420 million years. Nature Communications, 8, 14845, doi: 10.1038/ncomms14845.

As to the last ~1 Ma, perhaps he is not aware the numerous deep ice cores from low-accumulation regions of the East Antarctic Plateau which go back about 0.8 Ma, far further than the Vostok core to which he refers. A good starting point for reading would be the papers of

Edward J. Brook, and Christo Buizert, 2018: Antarctic and global climate history viewed from ice cores. Nature, 558, 200-208, doi: 10.1038/s41586-018-0172-5, and

Augustin, L. et al., 2004: Eight glacial cycles from an Antarctic ice core. Nature, 429, 623-628, doi: 10.1038/nature02599.

Regarding the author’s rebuttal comment of ‘… I wonder how the Reviewer knows what happened “in reality, during the Holocene” ’, suffice it to say that this period, starting after the finish of the LGM, is universally regarded as one of low variability in most climate parameters. As a particular aspect of this, Box 2 in Brook and Buizert’s paper shows very little CO2 variability over that period.

I was a little puzzled by the author’s response to my comment on the carbon budget. He comments that ‘… I respectfully disagree that the carbon budget could ever be balanced or close to it. In my view, balanced system is only a dead system, because it is the imbalances, whether small or big, that produce change, i.e., alive systems.’ In preindustrial times there were huge SEASONAL fluxes betewwnt the ocean, atmosphere and biosphere, but on an ANNUAL basis these averaged out to zero. However, it could not be argued that it was a ‘dead’ system.

The author’s presents the appropriate budget equation, i.e.,

Change of storage in the atmosphere = natural emissions – natural sinks + human emissions

From my comments above, if we are considering timescales of > 1 year, the first two terms on the right side of the equation will cancel. Hence the statement he makes ‘that each of natural emissions and natural sinks are more than 20 times greater than human emissions’ is (as I indicated before) meaningless for the overall trend of CO2 (and is only true – and irrelevant in the present context - if one is only considering the seasons). As I commented in my first review, examination of (and reference to) the ‘Keeling Curve’ (https://keelingcurve.ucsd.edu/) will help the author understand this.

The part of the text must be rewritten.

Once these points have been addressed in an appropriate manner I will be able to recommend acceptance.

Reviewer 2 Report

As the author's response to the first round of reviews was generally contrary and combative, and indeed "helped me to strengthen the paper against them" (referring to this reviewer in the Acknowledgements), I have no further comments to the author regarding this revision. I wish the author good luck in his retirement.

Reviewer 4 Report

I am proposing further procedure for publishing of the paper. I am satisfied with the answers on my review. 
